# LOX1- and PLP1-dependent transcriptional reprogramming is essential for injury-induced conidiophore development in a filamentous fungus

**Martín O. Camargo-Escalante,[1] Edgar Balcázar-López,[1] Exsal M. Albores Méndez,[2] Robert Winkler,[1] Alfredo Herrera-Estrella[1]**

**ABSTRACT** Fungi use oxylipins (oxidized lipids) as signaling molecules to induce asexual development. These molecules play an essential role in the response to wounding, exerting a protective effect against plant pathogens, and are part of the inflammatory process in animals. However, the physiological and molecular mechanisms triggered by oxylipins that lead to asexual development in fungi are not well understood. Using a genetic approach, mass spectrometry, and phenotypic analysis, we describe the functional role of a patatin-like phospholipase (*plp1*) and a unique lipoxygenase (*lox1*) in the response to injury in the model fungus *Trichoderma atroviride*. *lox1* and *plp1* are co-expressed and regulated by damage signaling and sensing components. Phenotypic analysis revealed an essential defect in the emergence of aerial hyphae in the *lox1* and *plp1* null mutant strains, blocking injury-induced conidiation. In addition, functional loss analysis demonstrated that both genes are essential for wound-associated linoleic acid-derived oxylipin 13-hydroxy-9Z,11E-octadecadienoic acid (13-HODE) production and the transcriptional reprogramming required for conidiation. *T. atroviride* requires LOX1 and PLP1 to induce transcription factors involved in asexual development such as *brlA*, *hox2*, and *azf1* homologs at the early stages of the response and at a later stage to activate lipid metabolism and the structural proteins involved in aerial mycelium emergence. Our study shows how the cooperative function of *lox1* and *plp1*, during the response to wounding, regulates the molecular and physiological processes of damaged-sensitized cells that lead to reproductive aerial mycelium development and consequently, ensure survival through asexual reproduction.

**IMPORTANCE** In addition to being considered a biocontrol agent, the fungus *Trichoderma atroviride* is a relevant model for studying mechanisms of response to injury conserved in plants and animals that opens a new landscape in relation to regeneration and cell differentiation mechanisms. Here, we reveal the co-functionality of a lipoxygenase and a patatin-like phospholipase co-expressed in response to wounding in fungi. This pair of enzymes produces oxidized lipids that can function as signaling molecules or oxidative stress signals that, in ascomycetes, induce asexual development. Furthermore, we determined that both genes participate in the regulation of the synthesis of 13-HODE and the establishment of the physiological responses necessary for the formation of reproductive aerial mycelium ultimately leading to asexual development. Our results suggest an injury-induced pathway to produce oxylipins and uncovered physiological mechanisms regulated by LOX1 and PLP1 to induce conidiation, opening new hypotheses for the novo regeneration mechanisms of filamentous fungi.

**KEYWORDS** *Trichoderma atroviride*, mechanical injury, lipoxygenase, patatin-like phospholipase, oxylipins, oxidized lipids, conidiation, aerial mycelium, transcriptome

Address correspondence to Alfredo Herrera-Estrella, alfredo.herrera@cinvestav.mx.

The authors declare no conflict of interest.

See the funding table on p. 19.

Fungi are sessile organisms capable of coping with the stress caused by mechanical injury through physiological and morphological mechanisms that warrant survival, such as sanitation, wound sealing, and hyphal regeneration. Like plants and animals, fungi use a complex molecular network comprised of signal transduction pathways, danger signals, and secondary signals that trigger transcriptional reprogramming (1) to detect and respond to damage. Due to the complexity of the mechanical damage response, the specific steps involved in hyphal regeneration and injury-induced conidiation (asexual reproduction) have not been elucidated. Previous studies in the filamentous fungus *Trichoderma atroviride* have shown that extracellular ATP (eATP) acts as a danger-associated molecular pattern (DAMP), signaling activation of Ca$^{2+}$ influx and release, which are required events for hyphal regeneration (2). In addition, injury-induced reactive oxygen species (ROS) production by the NADPH complex (Nox1/NoxR) activates the TMK3, a mitogen-activated protein kinase (MAPK) pathway essential for conidiophore development (2, 3).

In animals and plants, oxidized lipids play a central role in wound responses, regeneration, and immunity, acting as lipid mediators (4, 5). In common with ROS, endogenously produced oxidized lipids (oxylipins) function as oxidative agents and signaling molecules. However, their biological functions in the mechanical damage response in fungi are still unknown. These wound-induced signals are the product of the release of polyunsaturated fatty acids (PUFAs) by phospholipase A (PLA) and their subsequent oxidation by cyclooxygenases (COX) and lipoxygenases (LOX). In animals, COXs and LOXs metabolize arachidonic acid (AA) to produce eicosanoids such as prostaglandins and others that act as mediators of inflammatory and anti-inflammatory processes and activate the innate immune system (6).

Similarly, the plant lipoxygenase pathway metabolizes α-linolenic and linoleic acids into jasmonate and cyclopentenones, among other compounds, to establish a defense system analogous to the vertebrate innate immune system against insects and pathogens (7–9). However, in fungi, the participation of lipoxygenases and acyl-hydrolases to produce oxidized lipids in response to wounding has not been described. Interestingly, Hernández-Oñate et al. (3) reported that mechanical injury in *T. atroviride* leads to mRNA accumulation of a lipoxygenase (*lox*) and a patatin-like phospholipase (*plp*1) encoding genes, suggesting an essential role in the injury response. Thus, fungi would require this conserved mechanism to produce injury-induced oxylipins to respond to this stress.

Fungal lipoxygenases (LOXs) convert PUFAs into hydroperoxide derivatives, which act as precursors of a wide variety of oxylipins (10). However, the physiological role of fungal LOXs has yet to be understood. A previous study reported that a *lox1* null mutant of *T. atroviride* strain P1 produces fewer conidia than the wild-type (WT) strain in response to injury (11). Similarly, in *Aspergillus flavus* and *Aspergillus ochraceus,* a lipoxygenase participates in asexual development and mycotoxin production (12, 13).

The increase in intracellular oxylipin levels depends on PLA activity. Within the PLA superfamily, lipid acyl-hydrolases constitute a family of enzymes conserved in plants, animals, and fungi known as patatin-like phospholipases (14). In vertebrates, enzymes containing a patatin-domain such as Ca$^{2+}$-activated PLA$_2$s (cPLA$_2$) and Ca$^{2+}$-independent PLA$_2$B (iPLA$_2$B, PNLP9) regulate AA levels to produce eicosanoids and eliminate phospholipid peroxides avoiding cell death by ferroptosis (15–17). In addition, plant patatin-related PLAs (pPLA) modulate LOX-produced oxylipin levels and promote resistance to pathogens through cell death (18–20). Although oxidized lipids have biological activity in fungi, the control of their production by patatin-like phospholipases has yet to be studied.

Here, we describe the participation of *lox1* and *plp1* in injury-induced conidiation by promoting the emergence of aerial mycelium and regulating gene expression involved in asexual development. We further demonstrate the unprecedented functional epistasis between LOX and PLP1 in *T. atroviride*, which work together as in plants and animals, to produce the linoleic acid-derived oxidized lipid 13-hydroxy-9Z,11E-octadecadienoic acid

(13-HODE). Our study reveals the existence in fungi of a system analogous to those of plants and animals for the *lox-* and *plp1*-dependent production of an oxidized lipid with a biological function in response to injury that ensures the survival of *T. atroviride*.

## RESULTS

### Mechanisms for signaling and perception of mechanical damage regulate *lox1* and *plp1* in *T. atroviride*

A previous RNAseq analysis showed that two *T. atroviride* genes, encoding a unique lipoxygenase (LOX1) and one patatin-like phospholipase (PLP1), were induced and co-expressed within minutes in response to injury (3). This observation led us to hypothesize that LOX1 and PLP1 work together to produce oxylipins and participate in the response to mechanical damage. In search of mechanisms that regulate the expression of *lox1* and *plp1*, we analyzed the transcriptional behavior of both genes from the differential expression data of damage perception and signaling mechanisms previously established, such as the application of eATP and calcium and mutants (Δ*tmk1* and Δ*tmk3*) affected in MAPK-dependent signaling (2, 21). We observed that treatment with eATP, which induces conidiation (2), increased the levels of *plp1* and *lox1* mRNA, those of *lox1* being even higher than upon mechanical injury (Fig. S1A and B). By contrast, injury in the presence of BAPTA, a calcium-chelating agent that blocks regeneration and conidiation (21), decreased mRNA levels of both genes (Fig. S1C). In the Δ*tmk1* and Δ*tmk3* mutants, affecting regeneration and injury-induced conidiation (2), respectively, injury resulted in the upregulation of *plp1* and *lox1* but to lower levels than in the WT strain (Fig. S1D and E). Since these molecular components of the damage response regulate the expression of *lox1* and *plp1*, we hypothesized that the lost function of both genes negatively affects the regeneration capacity of hyphae and injury-induced conidiation.

To determine when these genes reach their maximum expression level and whether there was cross-regulation between them, we analyzed by time course the expression of *lox1* and *plp1* in the WT strain and the Δ*lox1* and Δ*plp1* mutants (Fig. S2). Reverse transcription-quantitative polymerase chain reaction (RT-qPCR) results showed that in the WT *lox1* increased its level of expression 30 min after injury (ai) as compared to an undamaged control, reaching a maximum at 90 min and decreased after 4 h (Fig. 1A). In comparison, *plp1* reached a maximum expression at 30 min, showing a decrease at 90 min and a further decrease at 4 h (Fig. 1A). The co-expression pattern of these two genes and their regulation by the same signaling and perception components suggested increased production of 13-hydroperoxy-9Z,11E-octadecadienoic acid (13-HpODE) or 9-hydroperoxy-10E,12Z-octadecadienoic acid (9-HpODE) derivatives as an oxidative stress response to damage at the level of membrane markers and signaling molecules.

### LOX1 and PLP1 work together to produce 13-HODE

Using in *silico* analyses, we classified LOX1 as a non-heme oxygenase that uses Fe as a cofactor and inferred that it oxidizes 13-C of linoleic acid (LA) to produce 13(S)-HpODE (Fig. S3 and S4). LOX1 is closely related to human 15-LOX and *Arabidopsis thaliana* 9- or 13-LOX. PLP1, on the other hand, is a functional small acyl hydrolase with a phosphate-binding motif related to a human PLP (PNPLA8) and three *A. thaliana* PLP (AtPLA-IVA and AtPLA-IVB and AtPLA-IIA) (Fig. S5).

To test the hypothesis that the *T. atroviride* LOX1 and PLP1 are functional and work together in oxylipin biosynthesis, we evaluated the oxidative metabolism of PUFAs at 30 and 60 min ai through targeted lipidomic analysis using liquid chromatography-mass spectrometry (LC-MS). Our results showed that ion 295.2272 m/z accumulated at lower levels in the mutants than in the WT (Fig. 1B), matching the theoretical elemental molecular formula $C_{18}H_{31}O_3$ [M-H]-. The identity of this ion was established as 13-HODE, a stable hydroxyl form of 13-HpODE, by comparing its collision-induced dissociation (CID) mass spectrum with databases (Fig. 1C). In the undamaged control condition, the

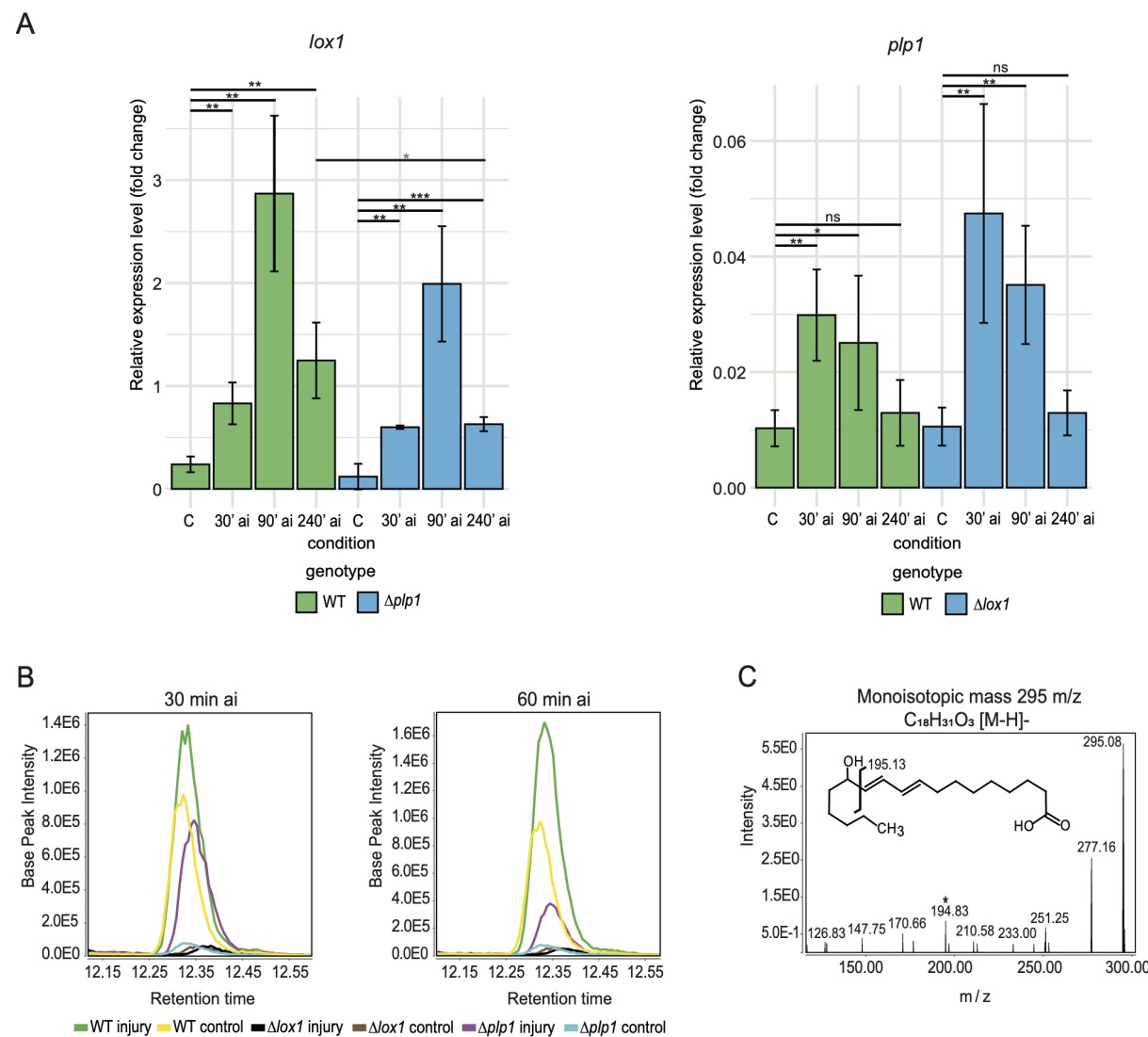

**FIG 1** *T. atroviride* responds to damage by inducing *lox1* and *plp1* to produce 13-HODE. (A) Time-course expression analysis of *lox1* and *plp1* by RT-qPCR in *WT* and knockout strains (*Δlox1* and *Δplp1*) relative to gene expression of *DNApolB*. Lines on bar plots indicate pairwise comparison in *t*-test analysis. Black asterisks indicate the statistical significance of the one-tail *t*-test performed: *$P < 0.1$; **$P < 0.05$; ***$P < 0.01$. Gray asterisks indicate the statistical significance ($P < 0.1$) of the two-tail *t*-test performed. Conversely, ns indicates no significant differences ($P > 0.1$). (B) Determination of 13-HODE by LC-MS in wound response at 30 and 60 min after injury. Fragmentation pattern of ion 295 at 25 eV.

WT strain produced 13-HODE with a peak intensity of $10^5$. After the injury, at 30 and 60 min, the intensity increased approximately threefold relative to the control in a time-dependent manner. As expected, we detected no signal of 13-HODE in *Δlox1* under control conditions or after injury. However, *Δplp*1 produced 13-HODE at 30 min ai in an order of magnitude of $10^5$, very similar to the levels observed in the WT under control conditions and with a decrease in intensity at 60 min ai (Fig. 1B). Although deletion of *plp1* does not affect the time-course expression of *lox1* (Fig. 1A), it does affect the production of 13-HODE after injury. We assumed that other active patatin-like phospholipases provide PUFAs during the injury response. There are five other patatin-like phospholipases encoded in the *T. atroviride* genome, and none of them show significant changes in expression level in response to injury (Fig. S6; Table S3). These results suggest that PLP1 and LOX1 are part of a pathway to produce 13-HpODE or 13-HODE in *T. atroviride*.

## A lipoxygenase and a patatin-like phospholipase participate in injury-induced conidiation but not regeneration

Using a cookie mold, we damaged mycelium to determine whether Δ*lox1* and Δ*plp*1 were affected by injury-induced conidiation. Our results showed that neither Δ*lox1* nor Δ*plp*1 conidiate in response to mechanical damage (Fig. 2A). However, in response to a pulse of blue light, the mutants produced conidia similarly to the WT strain (Fig. 2B), indicating that *lox1* and *plp1* are exclusively involved in injury-induced conidiation.

Given that *lox1* and *plp1* appear to regulate conidia production in response to mechanical damage, we evaluated this response at the morphological level in a time course. We determined whether Δ*lox1* and Δ*plp1* were affected by aerial mycelium induction or conidiophore development. Stereomicroscopy showed that Δ*lox1* and Δ*plp1* did not form aerial mycelium in the damaged area (Fig. 3A). At 4 h ai, there was no apparent difference between the WT and the mutants. However, aerial mycelium initiated a differentiation program at 6–8 h ai in the WT to form conidiophores with phialides in the injured zone. Unlike the WT, the mutants did not show the emergence of reproductive aerial mycelium in the damaged area. Quantification of conidia production induced by mechanical damage supported the observed defect in Δ*lox1* and Δ*plp1*

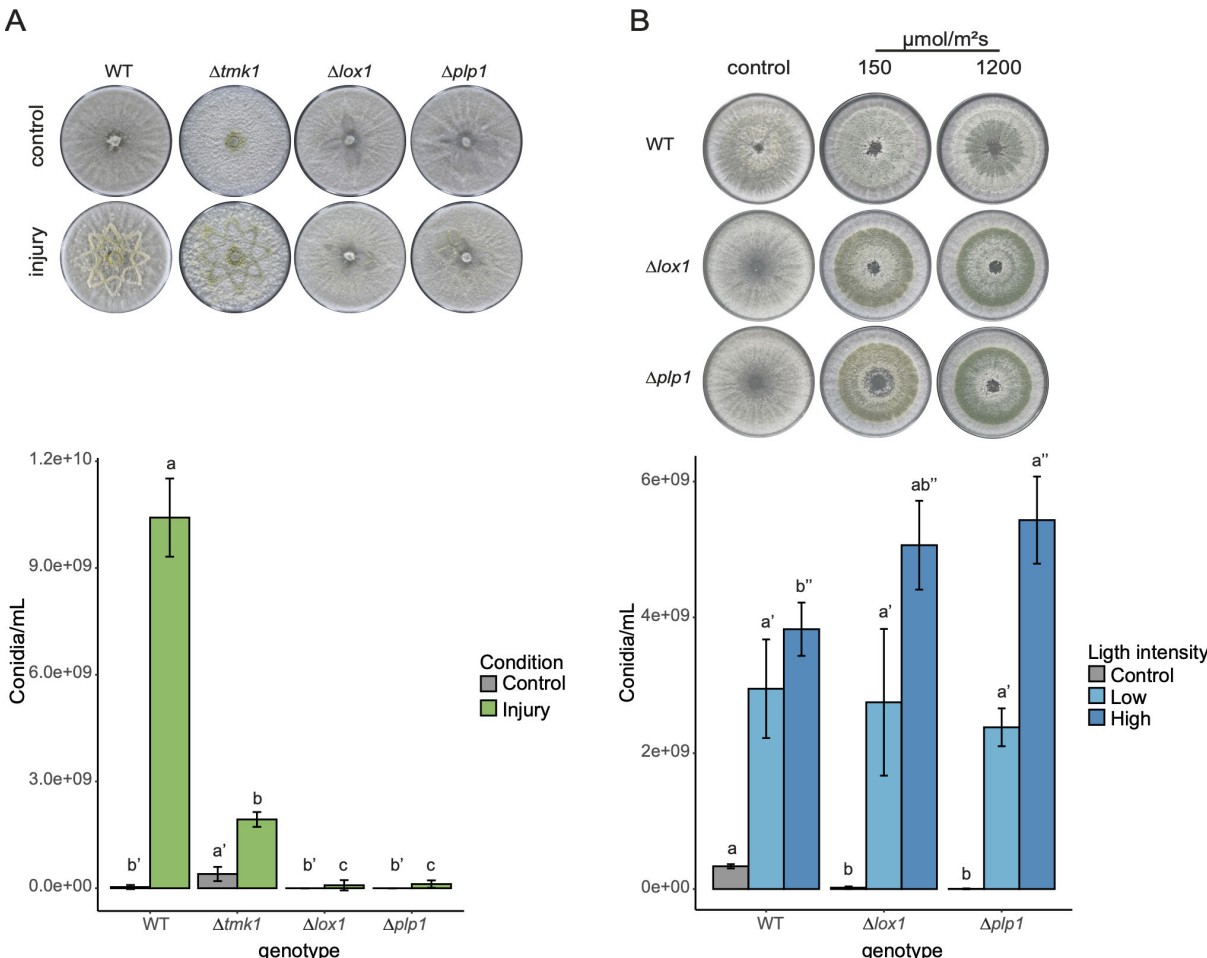

**FIG 2** *lox1* and *plp1* are necessary only for mechanical injury-induced conidiation. (A) Morphological response to mechanical damage and conidia number produced in WT and mutants incubated on PDA at 27°C during 48 h ai. Pictures correspond to representative images of injury-induced conidiation at 48 h after injury. (B) Conidiation by a blue light pulse at different light fluences. Colonies grown on PDA were exposed to an unsaturated (150 µmol/m²) and saturated (1,200 µmol/m²) blue light pulse and incubated at 27°C for 48 h. Representative photographs of blue light response on mycelium at 48 h after the stimulus. One-way analysis of variance and posterior test (Tukey HSD) were performed using an α = 0.05. Different letters above the bars indicate statistically significant differences.

strains (Fig. 3B; Fig. S7A). In conclusion, LOX1 and PLP1 are essential for aerial mycelium emergence, leading to phialide formation.

We further evaluated hyphal regeneration using a micro-culture system to determine whether the observed defect in injury-induced conidiation was due to reduced hyphal regeneration capacity. We quantified new hyphal tips and thin hyphae emergence near the rupture point of damaged hyphae over time (regeneration). We contrasted the hyphal regeneration capacity of Δlox1 and Δplp1 with that of the WT and Δtmk1, a mutant known to be affected by regeneration and injury-induced conidiation (Fig. 2A). Both mutants regenerated as efficiently as the WT strain (Fig. 3C; Fig. S7B). By contrast, Δtmk1 showed a 60% reduction in regeneration capacity compared to the WT. These results indicate that LOX1 and PLP1 play an important role in injury-induced conidiation but not regeneration.

## LOX1 and PLP1 co-regulate the transcriptional response to injury

To determine the possible consequences of the lack of PLP1 and LOX1 in the transcriptional landscape of *T. atroviride* in response to injury and identify the genes affected, we performed a transcriptome analysis at 90 min and 4 h ai (Fig. S8). First, we compared the transcriptional response of the Δlox1 and Δplp1 mutants against that of the wild-type strain. Multidimensional scaling (MDS) revealed that the mutants exhibit a similar expression pattern at the different response times evaluated, which sets them apart from the WT (Fig. 4A). We also noticed a distinct transcriptional response between the two times, which suggests two different morphological and physiological stages. From the differential gene expression analysis, we found that 46.24% and 51.67% of the genes with altered expression patterns relative to the WT at 90 min and 4 h ai, respectively, had the same behavior in the two mutants (Fig. 4B). Similarly, the mutants shared 61.22% of the genes with altered expression patterns in the non-injured control (Fig. 4B). These results indicate that in a significant proportion LOX1 and PLP1, regulate the expression of the same set of genes, both responsive and non-responsive to mechanical injury.

We detected 1,407 differentially expressed genes (DEGs) in the WT strain (608 up-regulated and 799 down-regulated) at 90 min ai and 714 genes (322 up-regulated and 392 down-regulated) at 4 h ai (Fig. S9; Data Set S1). By contrast, in Δlox1, we detected only 591 DEGs at 90 min and 448 genes at 4 h ai, and in Δplp1, 833 and 624 DEGs at 90 min and 4 h ai, respectively (Fig. S9; Data Set S1), indicating defects in the transcriptional reprogramming required at those stages of the response.

Given that LOX1 and PLP1 partially regulate the same transcriptional response, we identified 66 and 127 up-regulated genes at 90 min and 4 h ai, respectively, common to both mutants (Fig. 4C). This set of genes was over-represented in gene ontology (GO) terms of biological processes (BP) involved in cellular and redox homeostasis (P < 0.05). In the late response, the overrepresented GO terms were mainly carbohydrate metabolic processes for complex sugar hydrolysis and nitrate metabolic processes (Data Set S2). Of the down-regulated genes, the mutants shared 91 genes at 90 min ai (Fig. 4C), which were enriched in cell or subcellular component movement, unsaturated fatty acid, and phosphatidic acid biosynthetic processes, among others. Whereas at 4 h ai, they shared 49 genes overrepresenting mainly GO terms (P < 0. 05) related to the catabolic process of hydroxylated organic compounds, carbohydrate metabolic processes, and transmembrane transport (Data Set S2).

## Functional loss of *lox1* and *plp1* alters the transcriptional landscape of the response to injury

To obtain a broader view of the transcriptional response to injury, we carried out an enrichment analysis for each contrast (Fig. S10; Data Set S3). We found that the mutants responded early to oxidative stress (P < 0.01) caused by damage, in the same way as the WT, and the activation of homeostatic processes (P < 0.05) and cellular homeostasis [P < 0.01, false discovery rate (FDR) < 0.1] (Fig. 5A; Fig. S10). Among the genes involved in these processes, we found one encoding a calsequestrin (CASQ), which regulates

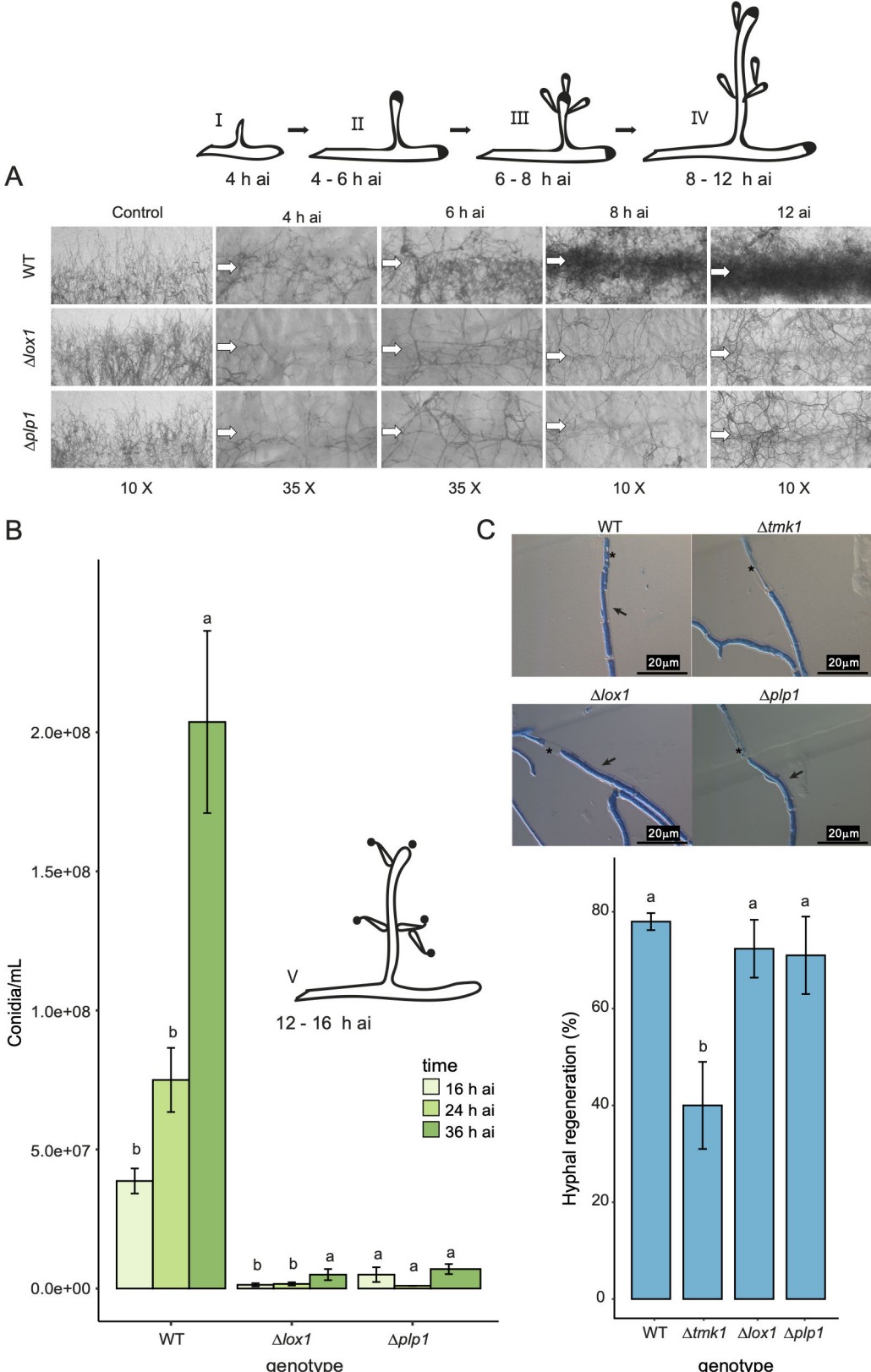

**FIG 3** Mechanical injury requires *lox1* and *plp1* to trigger aerial mycelium emergence and asexual development. (A) Stereomicroscopic view of aerial mycelium formation during the injury response (4 h, 6 h, 8 h, and 12 h ai). Arrows represent the damaged area by scalpel cut. (B) Time-course conidiation at 16 h, 24 h, and 36 h ai. The drawings represent the

**FIG 3** (Continued)

possible stages of development of the conidiophore: I and II aerial mycelium emerging; III and IV, aerial mycelium growing and conidiophore developing; V, conidiophore and conidia maturing. (C) Analysis of hypha regeneration at 90 min after injury. Pictures are representative images of regenerated hyphae at 90 min after stimuli. Arrows indicate tiny hyphae (regenerated hypha), except in *Δtmk1*, growing through cellular debris (*). Approximately 60–120 hyphae in the damaged area were visualized under the microscope, and the number of regenerating hyphae was counted. One-way analysis of variance and posterior test (Tukey HSD) were performed using $α = 0.05$. Different letters above the bars indicate statistically significant differences.

the calcium-induced calcium release system, with higher expression levels in *Δlox1* and *Δplp1* than in the WT (Fig. 5C). Similarly, one peroxidase (Tatro_001185-T1) and the thioredoxin encoding gene (*trx*) had higher expression in the mutants than in the WT (Fig. S11). The results also displayed a gene encoding a psiC factor-producing oxygenase (*ppo*C) homolog induced in *Δplp1* (Fig. 5A) and up-regulated in the WT at 90 min and 4 h (Fig. S12).

We also observed biological processes activated only in the WT ($P < 0.05$), represented in cluster 2 (Fig. S12), that could be important in damage-induced conidiation. A gene encoding a G protein-coupled receptor (GPCR) protein and the Ras GTPase-activating-like protein (RasGAP) encoding gene (Fig. 5C) (22) were up-regulated in the WT 90 min ai. In contrast with what was observed in the mutants. The Kyoto Encyclopedia of Gene and Genome (KEEG) enrichment analysis of up-regulated genes in the WT resulted in an over-representation ($P < 0.05$, FDR < 0.05) of the MAPK pathway and ribosome biogenesis, essential for regeneration and conidiation (Data Set S4).

In the late response, we found processes related to nitrogen metabolism ($P < 0.05$, FDR range <0.05 to 0.1) (Fig. 5B; Data Set S3) over-represented in the mutants. The relevant genes related to these functions, contained in clusters 1 and 3 (Fig. S12), encode an NADH-cytochrome b5 (nitrate reductase (NR) ortholog in fungi) and a nitrite reductase (NiR) that participate in nitric oxide (NO) production. Although damage induced these genes in the mutant backgrounds, at both stages of the response, the WT had expression levels of NR and NiR higher than the mutants (Fig. 5C). Conversely, in the WT, biological processes ($P < 0.01$, FDR range <0.05 to 0.1) that could be involved in asexual development, such as lipid metabolism, lipid biosynthetic processes, response to oxidative stress that continued induced, and monocarboxylic acid biosynthetic process (Fig. 5B) were over-represented. Whitin them, three differentially expressed genes in the WT, encode phosphatidyl serine decarboxylases (PSD), type I and II that perform the conversion of phosphatidylserine (PS) to phosphatidylethanolamine (PE) (Fig. 5C). As well as genes involved in terpene derivative production and fatty acid biosynthesis. The results of the KEGG enrichment analysis supported the observed representation of fatty acid and unsaturated fatty acid biosynthesis ($P < 0.05$, FDR range <0.05) (Data Set S4).

## Genes induced exclusively in the WT strain play an essential role in the transcriptional reprogramming during asexual development

Although the transcriptional response of the mutants to damage shared some genes with the WT strain, the mutations negatively affected the appropriate response to injury of many other genes, reflected in the induction or repression of necessary biological processes (Fig. 6A). These genes were called WT-unique genes (WUGs), at 90 min ai over-represented GO categories ($P < 0.05$, FDR range <0.05 to 0.1) related to the regulation of gene expression, cellular biosynthetic process, and regulation of macromolecule biosynthesis, among others. While at 4 h ai, WUGs represented categories ($P < 0.01$, FDR range <0.05 to 0.1) related to unsaturated fatty acid biosynthetic and metabolic processes and carboxylic acid biosynthetic processes (Data Set S5).

At the early stage of the response, we found that WUGs encoding several transcription factors (TFs) such as *crz1* and others belonging to the homeobox and bZIP family; membrane sensors such as a mechanosensitive ion channel family protein; MAPK pathway genes such as Tmk2, Tmk3 and Ste11, and genes encoding cell cycle proteins

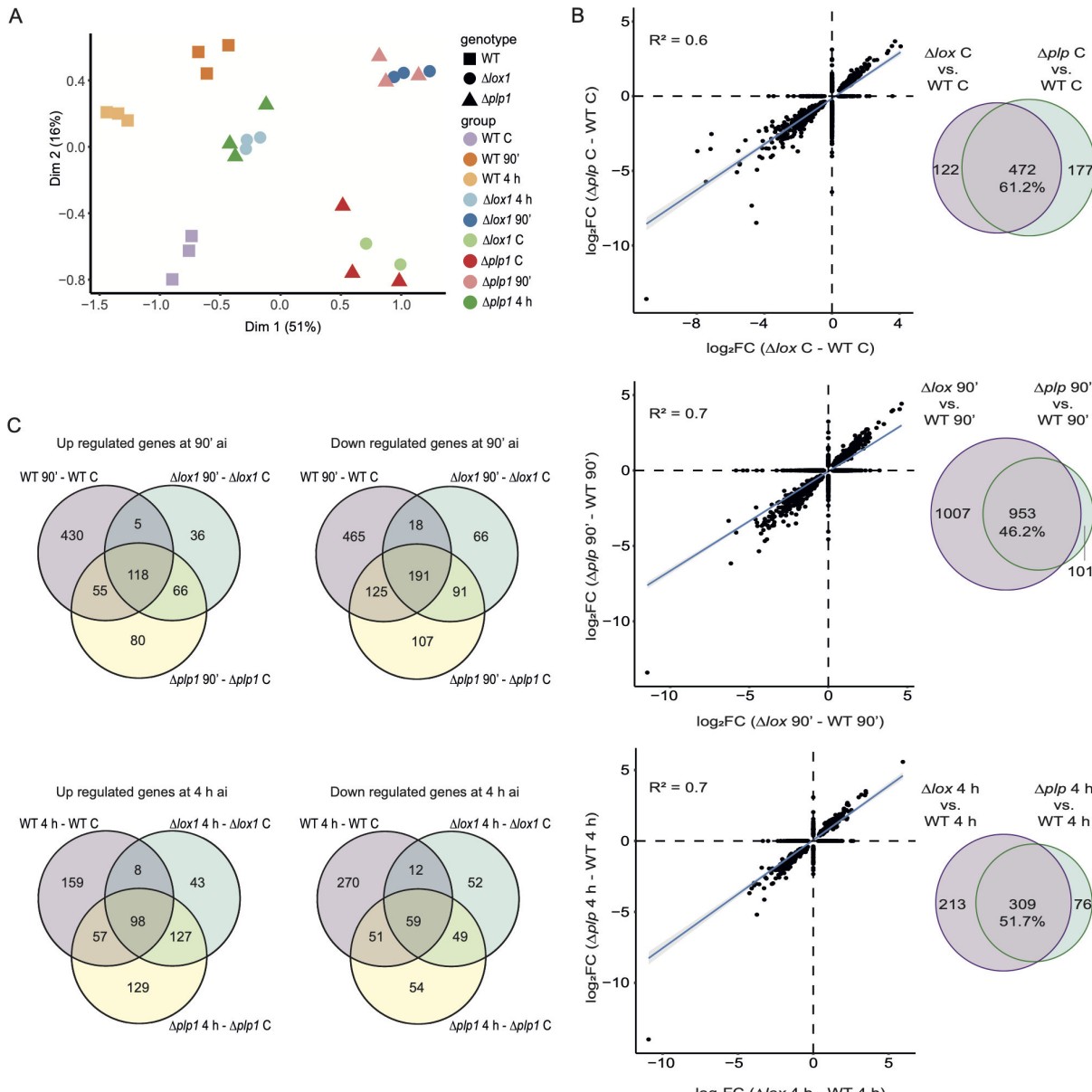

**FIG 4** *lox1* and *plp1* are affected in the regulation of a gene set in common. (A) Multidimensional scaling of normalized 500 features (genes); the axes indicate the percentage of explained variance. (B) Scatter plots of DEGs in Δ*lox1*-WT and Δ*plp1*-WT. DEGs of Δ*lox1*-WT a significantly strong association (P < 0.05 with Δ*plp1*-WT under control conditions and after injury. Grey areas indicate 95% confidence intervals. Venn diagrams of DEGs in the contrasts Δ*lox1*-WT and Δ*plp1*-WT in the control and at the indicated times after injury. (C) Venn diagram of up-regulated and down-regulated (log$_2$FC > |0.559|) in response to wounding in the WT, and the Δ*lox1 and Δplp1* mutant strains, results of Bayes quasi-likelihood F-test, LFC = 0.

like cyclin C were induced. During the late response stage, we found induction of WUGs participating in aerial mycelium development and asexual structure differentiation (Data Set S6).

To determine which induced WUGs were differentially expressed in the mutant backgrounds, we compared the transcriptional profile of Δ*lox1* and Δ*plp1* versus WT in each treatment (Data Set S7). We selected only the differentially expressed genes in the injury contrast in both mutants (Δ*lox1* injury-WT injury = Δ*plp1* injury-WT injury). These analyses yielded 82 selected genes in the injury contrast at 90 min and 26 genes at 4 h.

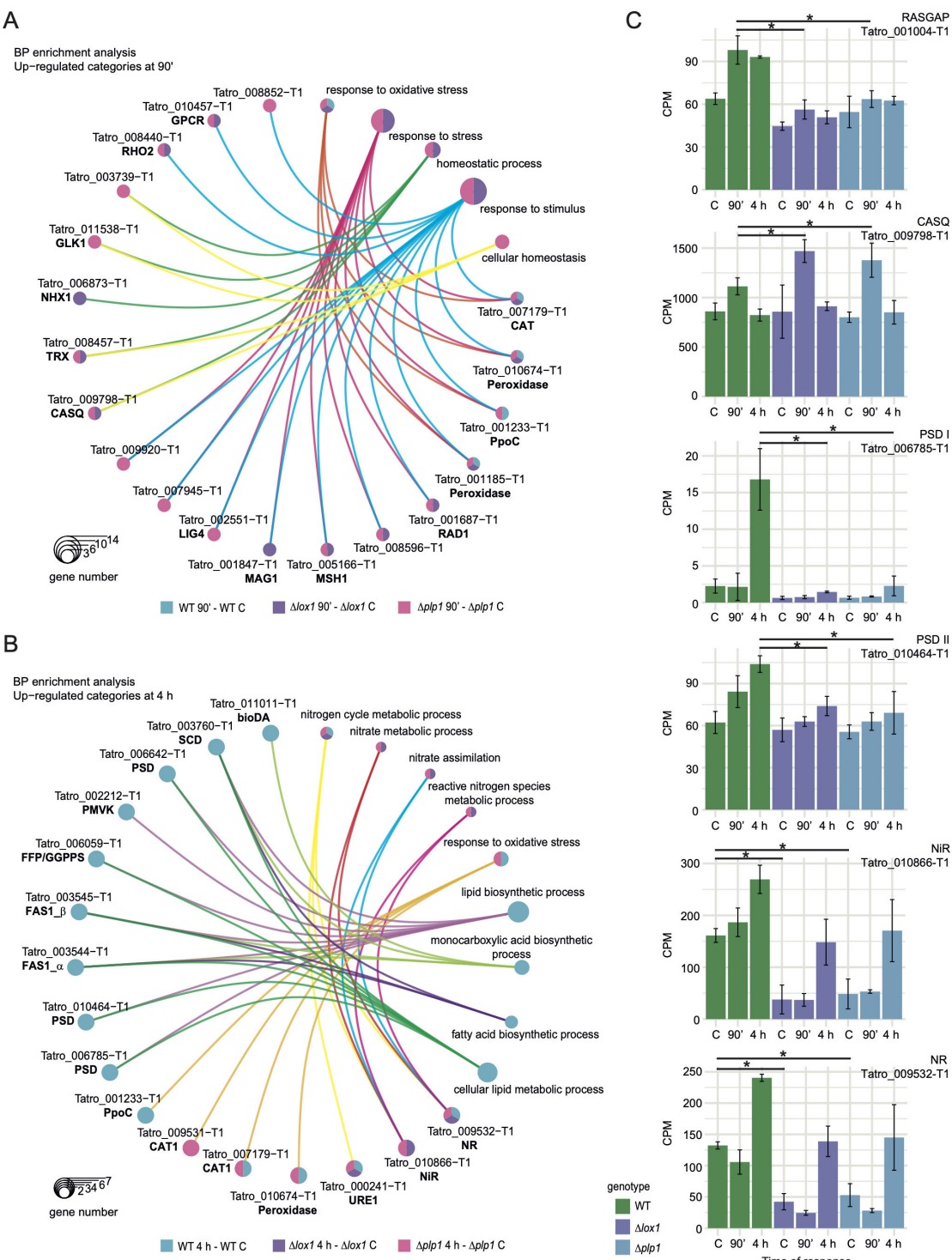

**FIG 5** Functional enrichment analysis of the differentially expressed genes in the mutants *lox1* and *plp1* and WT strains. (A and B) Linkage of up-regulated genes to biological processes overrepresented at 90 min and 4 h ai. Circle size indicates the number of genes in each biological process. (C) Count per million of representative transcripts in (A and B). Black lines on bar plots indicate pairwise comparison obtained from contrast *Δlox1*-WT and *Δplp1*-WT in control conditions and after injury. An asterisk indicates statistically significant differences, FDR < 0.05. Official NCBI gene names and symbols appear in bold.

In these analyses, three main groups of genes showed a reduced expression level in the mutant backgrounds in contrast with the WT at 90 min ai: Several ankyrin repeat protein and transcription factors, an $Mg^{2+}$ transporter (CorA), two WD40 repeat and two

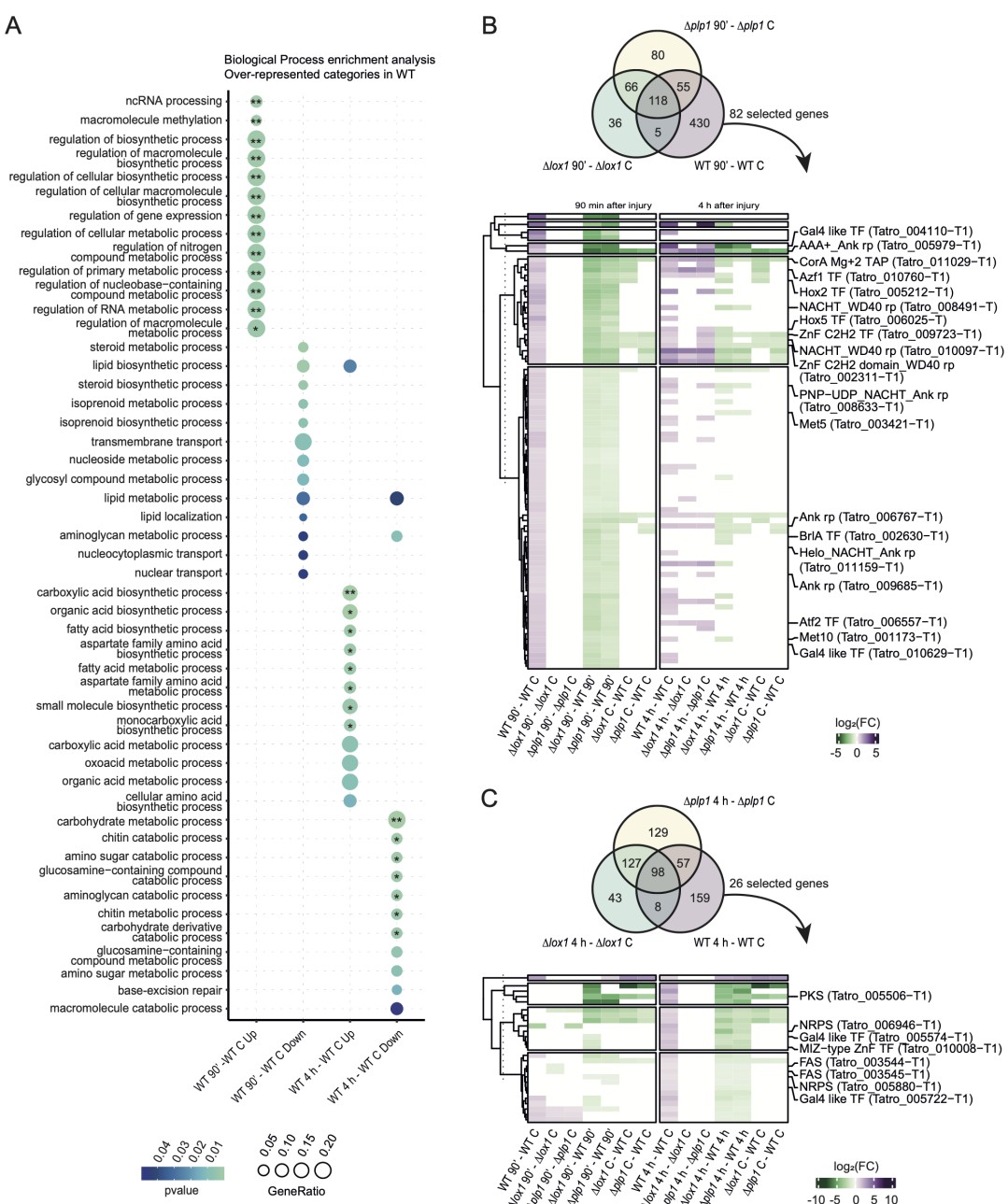

**FIG 6** Differential expression and functional enrichment analysis of genes induced exclusively in the *WT* (WUGs) in response to damage. (A) Over-representation of biological processes unique to the WT strain in injury (*P*-value < 0.01). Asterisks represent FDR adjusted *P*-value: **P* adjust <0.1; ***P* adjust <0.05. BP without asterisks indicates raw *P*-value < 0.05. (B) Heatmap of genes induced exclusively in the WT (induced WUGs) strain, selected according to differential gene expression in contrasts: *Δlox1* 90 min ai— *WT* 90 min ai and *Δplp1* 90 min ai—*WT* 90 min ai, and (C) *Δlox1* 4 h—*WT* 4 h ai and *Δplp1* 4 h—*WT* 4 h ai. The DEGs resulted from a Bayes quasi-likelihood F-test, LFC = 0 and FDR < 0.05.

sulfite reductases (Met5 and Met10) (Fig. S11). Among 10 ankyrin repeat proteins, for 7 of them the mutation drastically reduced their expression at both stages monitored (Fig. 6B). Similarly, induced WUGs associated with transcription factors had reduced expression levels in the mutants compared to the WT. At 90 min ai, these included two Cys2His2-type zinc finger proteins, one of which corresponds to the *Saccharomyces cerevisiae azf1* and the *Magnaporthe oryzae cos1* homolog (Table 1; Fig. 7). Similarly, a homeodomain (HD/Hox) family TF homolog of *M. oryzae* Hox5 (Fig. 7B; Table 1), two GAL-4 like TF, and a bZIP family TF, that has the closest correspondence to AtfB and AtfC

of *Aspergillus* species, had reduced expression in the mutants (Fig. 7; Table 1). At 4 h ai, transcription factor-associated WUGs included two GAL4-like TF and a MIZ-type zinc finger domain-containing TF homolog of the *S. cerevisiae* Nfi1, and WUGs related to fatty acid and phospholipid biosynthesis and natural product synthesis (Fig. 6C).

We also found a C2H2-type zinc finger TF and one with a homeobox domain expressed in the range from 1.6 to 3.5 lower fold than in the WT (Fig. 7). The homeo-domain family TF (Tatro_005212-T1) homolog of *Fusarium* species Htf1 and *M. oryzae* Hox2 is involved in phialide genesis (25, 26), while the C2H2 zinc finger TF is similar to *Purpureocillium lavendulum* BrlA (Table 1). Interestingly, we noted a *Botrytis cinerea* Atf1

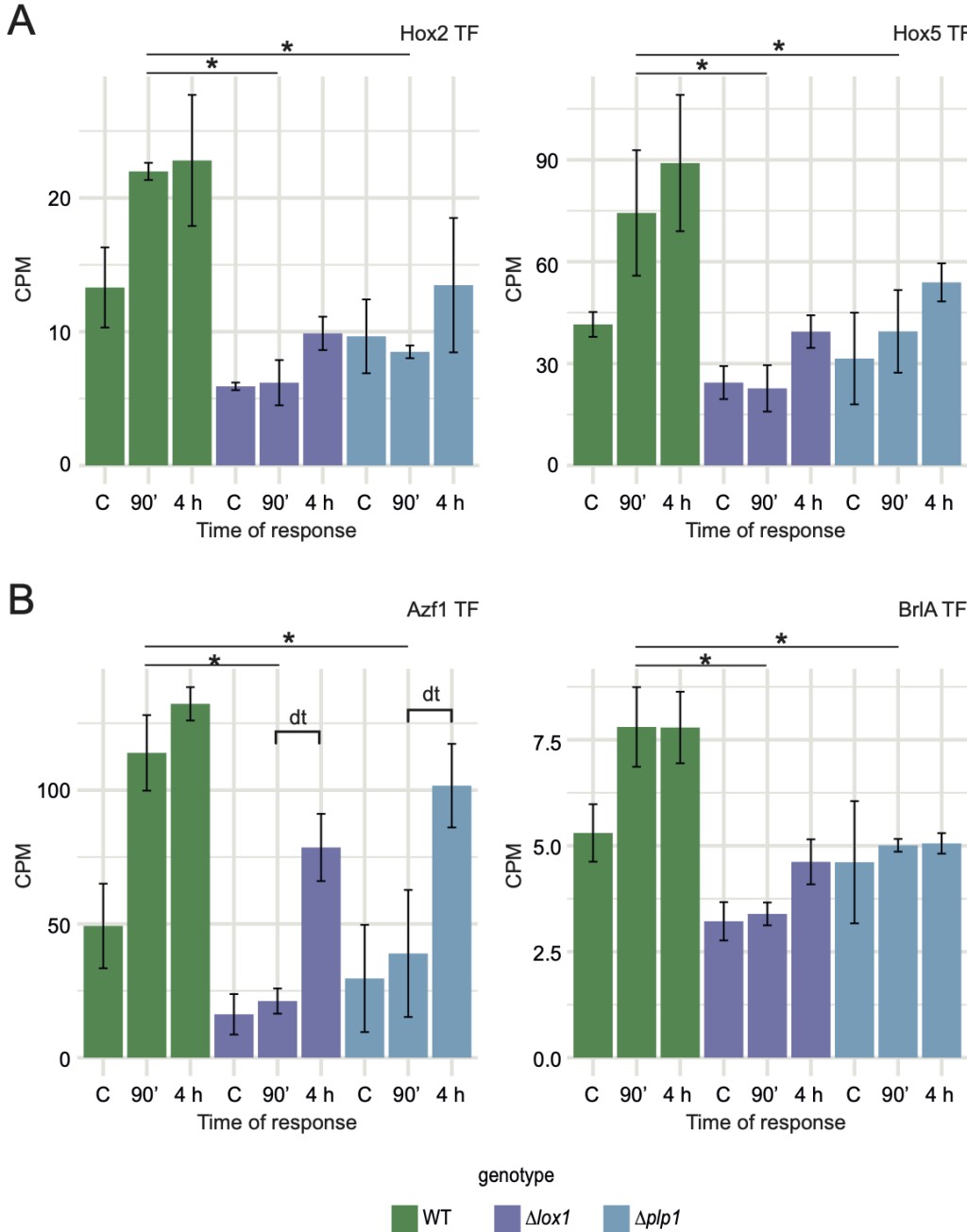

**FIG 7** Comparison of the expression levels in the mutant and WT strains of TFs involved in conidiation in ascomycetes. Counts per million (CPM) of the family of TFs homeobox (A) and C2H2 zinc finger (B). Black lines on bar plots indicate pairwise comparison obtained from contrast Δ*lox1*-WT and Δ*plp1*-WT subjected to injury and the control. An asterisk indicates FDR < 0.05, and a dt (delayed timing), delay in expression time.

**TABLE 1** *Trichoderma atroviride* transcription factors and their closet corresponding homologs of known biological function in other ascomycetes

| ID query | Subject sequence (accession number) | Organism | Query coverage | E-value | Identity (%) | Biological function |
|---|---|---|---|---|---|---|
| Tatro_010760-T1 | COS1 (KAH9432479) | *Pyricularia oryzae* | 43% | $3e^{-27}$ | 34.45% | Conidiophore stalk formation (23) |
| Tatro_002630-T1 | BrlA (AYJ71530) | *Purpureocillium lavendulum* | 96% | $6e^{-106}$ | 51.48% | Sporulation structure formation (24) |
| Tatro_006025-T1 | Hox5 (XP_003711331; MGG_07437) | *Pyricularia oryzae* | 55% | $1e^{-95}$ | 51% | Uncharacterized; K.O. sporulates (25) |
| Tatro_005212-T1 | Hox2 (XP_003718936; MGG_00184) | *Pyricularia oryzae* | 88% | $4e^{-35}$ | 30.36% | Phialide genesis (25, 26) |
| Tatro_006557-T1 | AtfC (XP_664453) | *Aspergillus nidulans* | 31% | $3e^{-12}$ | 32.17% | Heterodimers with AtfA; K.O. are sensitive to cell wall stress (*Aspergillus fumigatus*) (27) |
| [a]Tatro_006557-T1 | AtfB (XP_681912) | *Aspergillus nidulans* | 20% | $4e^{-10}$ | 35.06% | Heterodimer with AtfA (*Aspergillus fumigatus*); K.O. are sensitive to $H_2O_2$ (27–29) |

[a]Bidirectional Blast of amino acid sequence using PSI-Blast.
[b]K.O. = knockout strain.

homolog (Tatro_009572-T1) down-regulated in Δ*lox1* independently of the stimulus (Fig. S11).

## DISCUSSION

Within the well-known perception and signaling components of the damage response in *T. atroviride*, eATP perception is essential to detect pathogens and damaged tissues, leading to an oxylipin burst that induces defense mechanisms in plants and animals (6, 16, 30). Another fundamental component of damage signaling is calcium, whose intake blockage interrupts intracellular calcium release and downstream activation of calcium-dependent signal transduction pathways such as calmodulin and Crz1, blocking hyphal regeneration and induction of *lox1* and *plp1* (21). In *Trichoderma*, eATP increases calcium concentration in the cytoplasm (21), similarly to the P2 × 7 ligand-gate $Ca^{2+}$ channel that opens in response to eATP binding in animals. Although there is no known purinergic receptor in fungi, perception of eATP and $Ca^{2+}$ influx are essential for the induction of *lox1* and *plp1*.

Oxylipin production derives from the PUFA metabolism *via* the lipoxygenase pathway. This pathway comprises the LOX step to form lipid hydroperoxides and subsequent reactions to produce more complex oxylipins. Oxylipin levels depend on phospholipase A activity, such as patatin-like phospholipase activity. These enzymes regulate the lipoxygenase pathway releasing PUFAs from membrane phospholipids to serve as substrate for LOX. PLP1 and LOX1 homologs in animals and plants are connected to produce oxylipins: human PNPLP8 mediates leukotriene production and *A. thaliana* pPLA-IIA slightly regulates jasmonic acid production (20, 31). Based on our targeted lipidomic results, PLP1 and LOX1 work together to produce oxidized lipids. The lack of *lox1* resulted in no production of 13-HODE under any of the conditions tested in this study. By contrast, Δ*plp1* continued producing 13-HODE, although in minimal amounts. These results suggest that PLP1 mediates PUFA levels to produce 13-HODE *via* the LOX1 step.

Our results demonstrate that functional loss of LOX1 or PLP1 affected injury-induced conidiation but not hyphal regeneration. In this regard, both mutations repress aerial mycelium formation in the damage zone, and consequently, phialide-genesis does not occur. In agreement with our findings, it was recently described that in *T. atroviride* P1, LOX1 is involved in conidiation in darkness and mechanical injury (11). The absence of *lox*1/*lox*A in *Aspergillus* affects conidiation and aflatoxin production (12, 13). However, there are no reports on how lipoxygenase or its products regulate conidiation.

Our transcriptomic analysis showed that the *lox1* and *plp1* mutants displayed a similar transcriptional response, in agreement with the similarity of their phenotypes. Although there is no evidence on how LOX1 and PLP1 regulate the same set of genes, it could be through 13-HpODE or 13-HODE signaling. However, signaling through 13-HpODE or 13-HODE in fungi is unclear. In *A. nidulans* 13(S)-HpODE induces conidiation (32), which could be perceived through a GPCR (*grpD*) and generates a cAMP burst (33). In animals, 15-LOX can oxidize membrane phospholipids *via* direct oxidation of membrane PLs or indirect oxidation by the Land's remodeling pathway to bind to membrane receptors, alter membrane electronegativity, form pores, increase water permeabilization, and enhance the activity of GPCRs or function as a DAMP (34, 35). Interestingly, *T. atroviride* has an ortholog of *gprD* (Tatro_004209-T1) and an induced *gpcr* that could sense oxidized lipids. Therefore, we hypothesize that 13-HpODE/13-HODE signals *via* GPCRs.

Our transcriptional analysis showed that several processes related to asexual development and ROS scavenging, such as Ras GTPase activity, $Mg^{2+}$ transport, and sulfur availability, were activated by damage in the WT but not in the mutants. We found two RasGAPs and one RhoGAP encoding genes that decrease GTPase activity, expressed only in the WT in response to injury. In *Aspergillus* species, the levels of Ras GTPase activity dictate the type of aerial mycelium, where the gradual decrease in activity by RasGAP promotes asexual development (22, 36). Similarly, in *Fusarium graminearum*, RasGAP participates in conidiogenesis and the fusion of vesicles with the cytoplasmic membrane (37), which is vital for releasing cellular components such as signaling molecules, hydrophobins, aegerolysins, and enzymes with hydrolytic activity such as secretory phospholipase A2 and chitinases, among others, required during asexual development, oxidative stress tolerance, and defense.

Likewise, the response to damage induced a $Mg^{2+}$ transporter essential in processes such as cAMP formation during signal transduction by GPCRs and aerial mycelium formation in *M. oryzae* (38). In addition, two sulfite reductase encoding genes were up-regulated. These enzymes participate in S-adenosyl methionine (SAMe) and GSH biosynthesis to control gene expression through histone methylation, eliminate ROS and lipid peroxides, and biosynthesize phosphatidylcholine (39). In yeast, biosynthesis of phosphatidylcholine (PC) by SAMe and PE prevents SAMe from being used uncontrollably in histone methylation and ensures adequate gene expression according to the biological context (40). Beyond the biological processes involved in response to injury and conidiation, we also noted that induction of some NLR encoding genes involved in the innate immune response in fungi depends on LOX1 and PLP1.

During the early wounding response, several TFs of the C2H2, bZIP, and homeodomain type that play important roles in conidiation, as reported in other fungi, were induced only in the WT but not in the mutants. Our results showed two bZIP TFs homologs to ATF/CREB bZIP TFs crucial in the stress responses and conidiation, Atf1 and Atf2/AtfC. According to ROS signaling mechanisms reported in fungi, Atf1 is a downstream component of SakA/Sak1, the oxidative stress-activated MAPK involved in aerial mycelium differentiation, and the oxidative and osmotic stress response (28, 41, 42). In our model fungus, Tmk3, an ortholog of SakA/Sak1, is rapidly phosphorylated in a Nox1-dependent manner after injury. Atf2/AtfC can form a heterodimer with Atf1 to tolerate cell wall and oxidative stress (27, 29, 43). Our findings suggest that the oxidative stress caused by the oxidized lipids could induce Atf2/AtfC and Atf1. In this regard, in *B. cinerea*, oxidative stress induces BcAtf1 independently of BcSak1 (41). Therefore, we hypothesize that the oxidized lipids produced in the early stage of the response to injury induce Atf1 and Atf2/AtfC, and their active form depends on Tmk3.

Another TF induced only in the WT is the homeobox gene Tatro_005212-T1 (JGI: 164928) (*P*-value = 0.00744, FDR < 0.05), which has a homeodomain domain in its structure. Mutants of the homologs of Tatro_005212-T1 in *Fusarium* species (Htf1) and *M. oryzae* (Hox2) form conidiophores with aberrant phialides that do not produce conidia (25, 26).

Within the C2H2 zinc finger TFs, during the response, the *T. atroviride azf1* (Tatro_010760-T1), a homolog of the *T. reesei azf1* and *cos1* of *M. oryzae,* was induced. In *T. reesei*, an *azf1* mutant is delayed in conidiation (44). Furthermore, in *M. oryzae*, deletion of *cos1* results in the loss of conidiophore stalk formation and expression of fluffy genes involved in asexual development (23). Another C2H2 zinc finger TF up-regulated in response to wounding is the homolog of *blrA*. In *P. lavendulum*, Δ*brlA* mutants form conidiophore stalks but do not produce conidia (24, 45). Therefore, we suggest that these C2H2 zinc finger TFs in *T. atroviride* may be involved in aerial mycelium formation that differentiates into phialides and mature conidiophores in response to injury.

In contrast with the mutants, the WT strain continued responding to oxidative stress at the late stage. Ninety minutes after injury, we found up-regulation of genes related to biosynthetic and metabolic processes of cellular lipids, NO production, and oxylipin biosynthesis. Cellular lipid metabolism is vital for maintaining fluidity and integrity of the cytoplasm and organelle membranes and vesicle and lipid body formation. The latter have a functional role in aerial hyphae formation (stalk cell) since they store hydrophobins that help the erection of this structure (46). Our results also showed three PSD, one of type I and two of type II, responsible for converting PS in PE, the second type of phospholipid found in eukaryote cellular membranes. In fungi, type II PSDs localized in the Golgi, vacuoles, and endosomes are essential in asexual development. In *A. nidulans*, *psdB* mutants have morphological defects in vegetative hyphae and do not sporulate despite the PE content not being abated (47). In *F. graminearum*, a type II PDS (Fgpsd2) participates in sporulation, and its mutation results in significantly decreased lipid droplet formation (48). Although the role of PSDs in *T. atroviride* is unknown, PSDs maintain PE levels significant in the physiology, growth, and development of fungi.

Similarly, the transcriptional analysis showed over-represented reactive nitrogen species (RNS) metabolism and oxylipin biosynthesis. Two genes of the RNS metabolism encoding NiR (Tatro_010866-T1) and NR (Tatro_009532-T1), both with higher levels of expression in the WT than mutants at 4 hours ai, work synergistically to convert $NO_3$ in NO through oxidation of NAD(P)H as electron donor (49). In *A. nidulans*, NO synthesis depends mainly on NR, whose expression and NO levels increase during the first hour of conidiation, which entails the transcriptional induction of *brlA* to activate asexual development (50). In the case of oxylipins, the homolog of the *Aspergillus nidulans ppoC* increased its expression level ai only in the WT. In *Aspergilli*, *ppoC* positively regulates *blrA* expression to favor asexual rather than sexual development (51). Our transcriptome data showed that the *T. atroviride blrA* homolog (Tatro_002630-T1) was up-regulated at 90 min ai. Considering the evidence, the possible increased production of NO and the PpoC product during the emergence of aerial mycelium could explain the transcriptional behavior of *brlA*.

Our research has demonstrated two significant findings. First, regenerated hyphae require LOX1 and PLP1 to produce 13-HODE at significant levels. Second, LOX1 and PLP1 regulate injury-induced conidiation through transcriptional reprogramming that triggers activation of biochemical processes, induction of TFs, and structural proteins involved in aerial hyphae and asexual development in ascomycetes (Fig. 8). These results suggest that enzymatically oxidized lipids are a new input signal on damage-sensitized cells that induces injury-induced conidiation, acting downstream at different cellular levels and rapidly activating a transcriptional program required for asexual development. Thus, we propose that LOX1 and PLP1 work together to produce LOX-initiated oxidized PUFAs in fungi that induce asexual development to ensure survival in harmful environmental conditions.

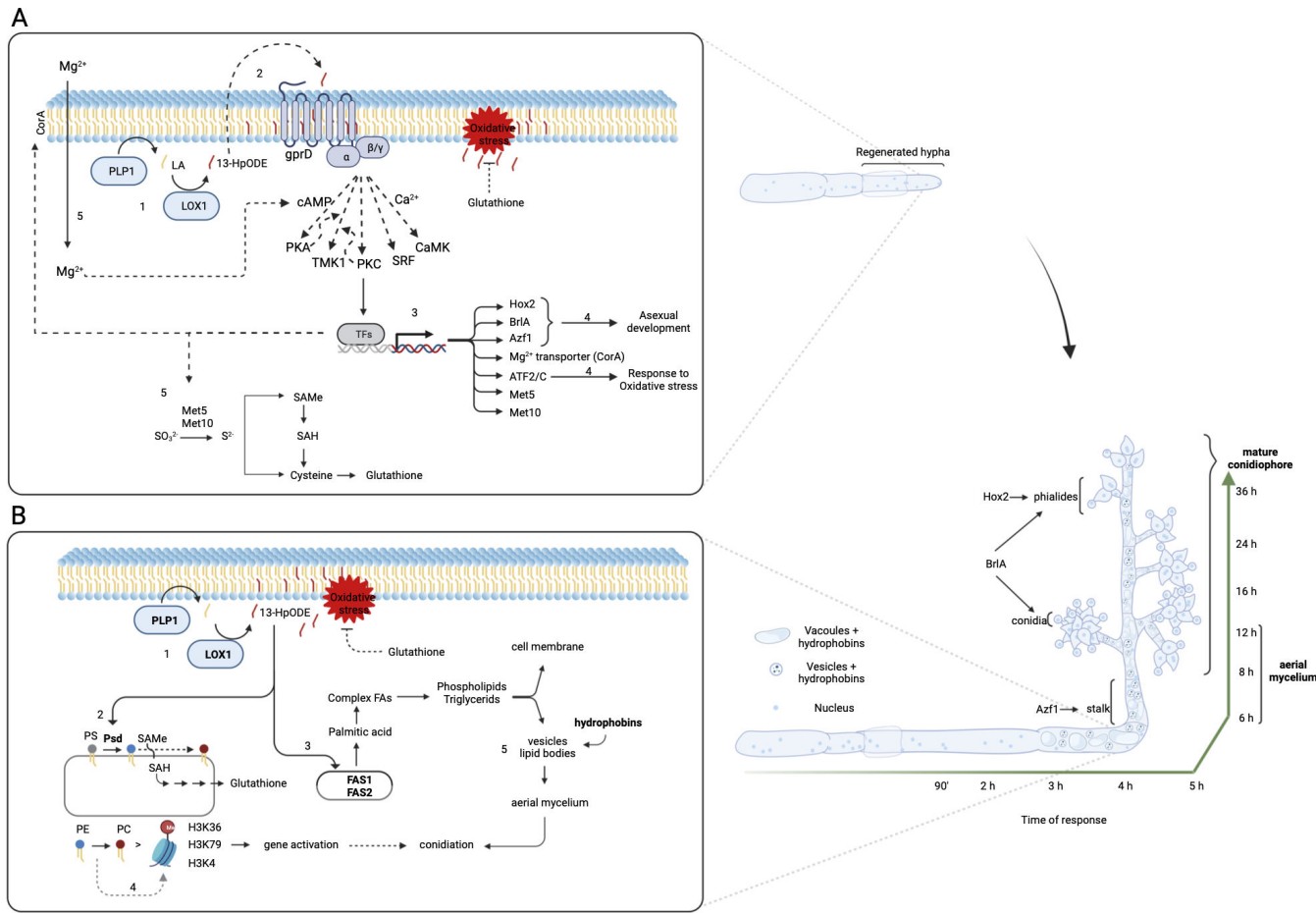

**FIG 8** Model of the role of *lox*1 and *plp*1 in the response to wounding. (A) In the early stage of the response (30–90 min after injury), Lox1 and Plp1 produce 13-HpODE (1), which could be sensed by GPCR or incorporated in membrane phospholipids (2), leading to GPCR activation. GPCR activation would, in turn, activate a variety of signaling cascades, including that of the MAPK Tmk1 and the calcium-dependent CaMK, to activate TFs, leading to essential transcriptional reprogramming (3). Modifying the transcriptional landscape would trigger the oxidative stress response (Atf2/C) and asexual development such as BrlA, Hox2, and Aztf1 involved in stalk formation and phialide development where aerial mycelium emerges (4). In addition, genes involved in the biochemical and physiological process, such as CorA (5) that positively feedback cAMP burst and helps transcription, translation, and DNA replication fidelity process, and Met5 and Met10 (6) that reduce $SO_3^{2-}$ to $S^{2-}$ to produce SAMe and glutathione, which could counteract oxidative stress provoked by oxidized lipids and ROS and use SAMe for histone methylation and phosphatidylcholine (PC) formation. (B) At the late stage, the response depends on LOX1 and PLP1 (1), which induce PSDs and FASs involved in phospholipid (2) and lipid biosynthesis (3). PSDs convert phosphatidylserine (PS) to phosphatidylethanolamine (PE). The latter could end in PC using SAMe as a methyl radical supplier. In this regard, PE could work as a sink, preferring PC production instead of histone methylation, which is involved in conidiation in fungi (4; gray dashed arrow). FASs produce palmitic acid for building complex fatty acids and, therefore, phospholipids and triglycerides to form lipid bodies, vesicles, and vacuoles that, in conjunction with hydrophobins (5), help with the appearance and erection of the reproductive aerial mycelium.

## MATERIALS AND METHODS

### Strains and culture conditions for injury-induced conidiation, light response, and growth experiments

The *T. atroviride* IMI206040 strain was used as the WT strain. The *Δtmk1* and *Δtmk3* knockout strains were reported previously (21). The DNA sequence of *lox1* (JGI: 33350) and *plp1* (JGI: 28577) was downloaded from *T. atroviride* genome v2.0 (https://myco-cosm.jgi.doe.gov/Triat2/Triat2.home.html). *Δlox1* and *Δplp1* mutants were generated and analyzed in this study. All strains were propagated in potato dextrose agar (PDA; Difco) at 27°C. For stress response (conidiation by injury and light) and growth experiments, we used 150 mm Petri dishes containing 30 mL of PDA. For obtaining vegetative mycelium, conidia were inoculated, and the fungus was allowed to grow for 36 h at

27°C in complete darkness, using aluminum foil as a wrap. We inoculated 100 conidia in glass slides covered with 2 mL of PDA for hyphal regeneration experiments and incubated them at 27°C. The mycelium was harvested using cellophane sheets overlaid on 150 mm Petri dishes containing PDA for experiments that required metabolite and RNA extraction.

## Injury-induced conidiation and hyphal regeneration

Thirty-six-h-old vegetative mycelium, obtained as described above, was damaged using a cookie mold in a dark room under a red security light and incubated for 48 h at 27°C. Conidia were harvested using 5 mL sterile water and quantified in a Neubauer chamber. For hyphal regeneration experiments, 50–100 fresh conidia were inoculated at the center of a microculture system, incubating for 18–20 h at 27°C in complete darkness. Vegetative mycelium was damaged with a sterile scalpel. After damage, mycelium was incubated at 27°C and stained with lactophenol blue at 60, 90, and 120 min after injury (ai). Mycelia were observed on a Leica DM6000-B microscope fitted with a 100× objective HCX PL Fluotar (0.75 N.A.) and photographed with a Leica DFC 420C camera.

## Light-induced conidiation

To evaluate blue light-induced conidiation, $1 \times 10^6$ conidia in a 10 µL drop were inoculated on PDA medium at the center of the Petri dish and allowed to grow for 36 h in complete darkness at 27°C. Finally, the colony was exposed to non-saturating (150 µmol/m$^2$) and saturating (1,200 µmol/m$^2$) light fluences and incubated for 48 h at 27°C in the dark. Conidia were harvested using 5 mL of sterile water and counted using a Neubauer chamber.

## Reverse transcription quantitative polymerase chain reaction analysis of *lox1* and *plp1* expression

Mycelia of the *Δlox*, *Δplp1*, and WT strains were collected 30, 90, and 240 min after injury and their corresponding undamaged control. Total RNA was extracted with TRIzol (Invitrogen). RNA samples (1 µg) were reverse-transcribed using oligo-dT (18) priming with Superscript II (Invitrogen). The first-strand cDNA produced was used as a template in quantitative PCR (qPCR) experiments, for which a set of primers was designed to amplify each gene under study (Table S1) specifically. The reaction mixture and PCR program were carried out as described (21). Transcript levels of target genes were normalized against the level of the *DNA polymerase family B* gene (JGI: 53190) using the $2^{-\Delta CT}$ relative quantification method (52).

## Mutant generation using doublejoint PCR and confirmation

Using the double-joint PCR method, the open reading frames (ORF) of the *lox1* and *plp1* genes were replaced by a hygromycin resistance cassette (*hph*) as previously described (53). Primers were designed for ORF replacement by homologous recombination (Table S1). These constructs were used for the PEG-mediated protoplast transformation of the WT strain (54). Five single spore isolation rounds in media containing hygromycin were carried out before confirmation by Southern blot.

## Phylogenetic analysis

Amino acid sequences were obtained by BLAST using LOX as a query. Lipoxygenase sequences were aligned using ClustalW in The GUIDANCE2 Server (http://guidance.tau.ac.il/) and subjected to Bayesian analysis applying the mixed amino acid substitution models. The analysis was performed with Mr. Bayes using one cold chain that performed two million generations in total. Sampling trees delivered Bayesian posterior probabilities (PP) generated every 500 generations.

## Oxylipin analysis by LC-MS

For lipidomic analysis and oxylipin determination, 30 mg of lyophilized mycelium was covered with 500 µL MeOH with 0.0025% (wt/vol) of butylated hydroxytoluene and homogenized by vortex for 2 min and sonicated 20 min. Next, the mixture was cooled at −80°C for 10 min for protein precipitation. Then, 400 µL diluted HCl (pH 3) was added to the raw methanol extract. The methanol/water phase was extracted with $CH_2Cl_2$ (2 × 1 mL). After phase separation by centrifugation, dichloromethane extracts were combined and dried with $N_2$. The extracts were resuspended in 100 µL of MeOH and stored at −80°C. The samples were analyzed by LC-MS, using a Dionex UltiMate 3000 HPLC (Thermo Scientific, Waltham, MA, USA) coupled to an Orbitrap Fusion Tribrid Mass Spectrometer (Thermo Scientific) with an electrospray source. To separate metabolites, we used an AccuCore C18 column (4.6 × 150 mm) with 2.6 um size particles as stationary phase and 0.5% acetic acid in LC-MS grade water (solvent A) and 0.5% acetic acid in LC-MS grade acetonitrile (solvent B) as mobile phase, employing at a flow rate of 0.5 mL/min and the following 18 min gradient: 0 min, 1% solvent B; 3 min, 15% solvent B; 6 min, 50% solvent B; 10 min, 90% solvent B; 12 min, 90% solvent B; 14 min, 50% solvent B; 16 min, 15% solvent B; and 18 min, 1% solvent B. The column temperature was controlled at 40°C, and the injection volume was 10 µL. The full MS spectra were acquired in negative and positive modes at 3, 500 volts, ranging from 50 to 2,000 m/z. Oxidized lipids were identified using higher-energy collisional dissociation (CID) with energy from 25 to 30 v. 13-HODE was determined using the mass transition 295→195. Samples were analyzed using MZmine, and the peak area was normalized employing the average intensity area.

## RNAseq and differential expression analysis

We collected mycelium of the WT, Δlox1, and Δplp1 at 0 h (undamaged control), 90 min, and 4 h after injury. Then, harvested mycelia were used for total RNA isolation (Invitrogen). The quality and quantity of the total RNA were determined using an RNA 6000 Nano Chip run in an Agilent Bioanalyzer platform. Each RNAseq library was generated using the MGIEasy RNA library Prep Set protocol and later sequenced in a pair-end 150 bp format in the MGISEQ-2000 sequencing platform with DNBSEQ technology. Each library produced, on average, 9.7 million raw reads. To obtain a read count matrix, we filtered reads with FastQC (V. 0.11.9) for mapping to the *T. atroviride* genome using HISAT2 (V. 2.2.1) and counted the reads overlapping each gene with HTseq (V 2.0.2) (Table S2).

We used the *edgeR* (V. 3.36.0) package in R for differential gene expression analyses and fitted data to a generalized linear model (GLM). In each comparison, we determined the differential expression using a Bayes quasi-likelihood F-test to a specified log2-fold change (LFC) threshold, LFC = 0, and we considered differentially expressed genes with an FDR < 0.05. The genes that responded to wounding were filtered using absolute LFC > 1.47.

For enrichment analysis of GO terms and The KEGG pathways, we used the *clusterProfiler* (V. 4.2.2) package in R. We considered enriched GO terms those that had a $P < 0.05$ and significantly enriched GO terms and KEGG pathways those that had a $P < 0.05$ and an FDR in the range <0.05 to 0.01. For hierarchical clustering groups, we filtered GO terms using only $P < 0.01$. The protein domain, functional annotation, orthology, and KO (KEGG Orthology) were predicted in eggnog-mapper v2 (http://eggnog-mapper.embl.de/), introducing the amino acid sequence encoding in each gene model. The scripts used for the enrichment analysis in this work are available at https://github.com/OrlanC/cp_enrichment.

## Statistical analysis

Statistical analysis was performed in R version 4.1.2. Phenotypic experiments were analyzed using an one-way analysis of variance and Tukey's test for multiple comparisons

with α = 0.05 and confidence level of interval (CI) = 0.95. qPCR results were analyzed using one-tailed and two-tailed $t$-tests with α = 0.05 and 0.1 and CI = 0.95 and 0.90. Data were represented as mean ± standard deviation (SD) from three biological replicates.

## ACKNOWLEDGMENTS

We thank Pedro Martínez-Hernández, Maria I. Cristina Elizarraraz Anaya, and María Teresa Carrillo Raya for their technical assistance. We also wish to thank Dr. June Simpson for her critical reading of the manuscript.

M.O.C.-E. is indebted to Consejo Nacional de Ciencia y Tecnología (CONACYT) for a doctoral fellowship (564172). This work was supported by CONACYT grant—Investigación en Fronteras de la Ciencia (FON.INST./117/2016) to A.H.-E.

## AUTHOR AFFILIATIONS

[1]Laboratorio Nacional de Genómica para la Biodiversidad-Unidad de Genómica Avanzada, Cinvestav, Irapuato, Guanajuato, Mexico
[2]Escuela Militar de Graduados de Sanidad, Universidad del Ejército y Fuerza Aérea Mexicanos, Secretaría de la Defensa Nacional, Mexico City, Mexico

## PRESENT ADDRESS

Edgar Balcázar-López, Departamento de Farmacobiología, Centro Universitario de Ciencias Exactas e Ingenierias, Universidad de Guadalajara, Guadalajara, Mexico

## AUTHOR ORCIDs

Martín O. Camargo-Escalante  http://orcid.org/0000-0003-2271-236X
Edgar Balcázar-López  http://orcid.org/0000-0003-0616-309X
Exsal M. Albores Méndez  http://orcid.org/0000-0002-0813-3428
Robert Winkler  http://orcid.org/0000-0001-6732-1958
Alfredo Herrera-Estrella  http://orcid.org/0000-0002-4589-6870

## FUNDING

| Funder | Grant(s) | Author(s) |
| --- | --- | --- |
| Consejo Nacional de Ciencia y Tecnología (CONACYT) | FON.INST./117/2016, Doctoral Fellowship 564172 | Alfredo Herrera-Estrella |
| | | Martín O. Camargo-Escalante |

## AUTHOR CONTRIBUTIONS

Martín O. Camargo-Escalante, Data curation, Formal analysis, Investigation, Methodology, Writing – original draft, Writing – review and editing | Edgar Balcázar-López, Investigation, Methodology, Writing – review and editing | Exsal M. Albores Méndez, Methodology, Writing – review and editing | Robert Winkler, Investigation, Methodology, Supervision, Writing – review and editing | Alfredo Herrera-Estrella, Conceptualization, Funding acquisition, Methodology, Resources, Supervision, Writing – original draft, Writing – review and editing

## DATA AVAILABILITY

All RNA-seq data relevant to this publication are available from the NCBI's Gene Expression Omnibus database (GSE235273; *lox1* and *plp1* RNA-seq).

## ADDITIONAL FILES

The following material is available online.

## Supplemental Material

**Data Set S1 (Spectrum02607-23-s0001.xlsx).** Supplementary Data Set 1. Differentially expressed genes (DEGs) and their annotation in the comparison WT injury - WT control, *Δlox1* injury - *Δlox1* control, *Δplp1* injury - *Δplp1* control.

**Data Set S2 (Supplementary Data Set 2.xlsx).** Supplementary Data Set 2. Enrichment analysis of Biological Process GO terms shared between *Δlox1* injury - *Δlox1* and *Δplp1* injury - *Δplp1* control.

**Data Set S3 (Spectrum02607-23-s0003.xlsx).** Supplementary Data Set 3. Enrichment analysis of Biological Process GO terms in the contrasts WT injury - WT control, *Δlox1* injury - *Δlox1* control, *Δplp1* injury - *Δplp1* control.

**Data Set S4 (Spectrum02607-23-s0004.xlsx).** Supplementary Data Set 4. Enrichment analysis of KEGG pathways in the contrasts WT injury-WT control, *Δlox1* injury-*Δlox1* control, *Δplp1* injury-*Δplp1* control.

**Data Set S5 (Spectrum02607-23-s0005.xlsx).** Supplementary Data Set 5. Enrichment analysis of Biological Process GO terms of WT-unique genes (WUGs).

**Data Set S6 (Spectrum02607-23-s0006.xlsx).** Supplementary Data Set 6. List of up-regulated WT-unique genes (WT injury - WT control).

**Data Set S7 (Spectrum02607-23-s0007.xlsx).** Supplementary Data Set 7. Differentially expressed genes (DEGs) in the comparison *Δlox1* - WT and *Δplp1* - WT in each condition.

**Supplemental figures and tables (Spectrum02607-23-s0008.pdf).** The file contains 12 supplemental figures, three supplemental tables and expanded methods.

## Open Peer Review

**PEER REVIEW HISTORY (review-history.pdf).** An accounting of the reviewer comments and feedback.

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
