## [Reviewer comments · Microbiology Spectrum]

Microbiology Spectrum

LOX1 and PLP1-dependent transcriptional reprogramming is essential for injury-induced conidiophore development in a filamentous fungus

Martín Camargo-Escalante, Edgar Balcázar-López, Exsal Albores-Méndez, Robert Winkler, and Alfredo Herrera-Estrella

Corresponding Author(s): Alfredo Herrera-Estrella, Centro de Investigacion y de Estudios Avanzados del Instituto Politecnico Nacional

Review Timeline:

Submission Date:	June 21, 2023
Editorial Decision:	August 18, 2023
Revision Received:	September 22, 2023
Accepted:	October 4, 2023

Editor: Lea Atanasova

Reviewer(s): The reviewers have opted to remain anonymous.

Transaction Report:

DOI: <https://doi.org/10.1128/spectrum.02607-23>

August 18, 2023

Dr. Alfredo H Herrera-Estrella
Centro de Investigacion y de Estudios Avanzados del Instituto Politecnico Nacional
National Laboratory of Genomics for Biodiversity-Advanced Genomics Unit
Km. 9.6 Libramiento Norte Carretera Irapuato-Leon
Irapuato, Guanajuato 36824
Mexico

Re: Spectrum02607-23 (LOX1 and PLP1 dependent transcriptional reprogramming is essential for injury induced conidiophore development in a filamentous fungus)

Dear Dr. Alfredo H Herrera-Estrella:

Thank you for submitting your manuscript to Microbiology Spectrum. Based on the reviewers comments you are kindly invited to submit a revised version of your manuscript. One of the reviewer specifically requested to improve the English language. When submitting the revised version of your paper, please provide (1) point-by-point responses to the issues raised by the reviewers as file type "Response to Reviewers," not in your cover letter, and (2) a PDF file that indicates the changes from the original submission (by highlighting or underlining the changes) as file type "Marked Up Manuscript - For Review Only". Please use this link to submit your revised manuscript - we strongly recommend that you submit your paper within the next 60 days or reach out to me. Detailed instructions on submitting your revised paper are below.

Link Not Available

Sincerely,

Lea Atanasova

Journals Department
Reviewer comments:

Reviewer #1 (Public repository details (Required)):

In the manuscript is written that that all the RNAseq data relevant for this analysis are available at NCBI omnibus. However the link, for understandable reason, is protected till defined date. But how are the reviewers able to verify the analysis if we do not have access to the data? Moreover, at the beginning of the results section, the authors refer to a previous RNAseq analysis, from a 2012 publication. When I tried to access those dataset, the link were giving an error. Of course this is not the authors fault, but considering some of the authors are the same between the two publication... it just doesn't give me a good feeling not being able to verify the original data

Reviewer #1 (Comments for the Author):

The manuscript "LOX1 and PLP1 dependent transcriptional reprogramming is essential for injury induced conidiophore development in a filamentous fungus" is a well written work, providing a lot of information and data regarding mechanical injury response in *Trichoderma atroviride*. Initially I identified only minor points that could be slightly changed to improve the manuscript. At a deeper look, however, some more relevant concerns were also raised.

Minor concerns:

- Line 112: 13-HODE the explicit form of the abbreviation is only written in the abstract. Wouldn't be more correct to have it at least once in the main text?
- Line 130: The sentence starting with "While...", perhaps style choice, but just the fact that it started with that made the sentence difficult for me to understand at a first reading. Maybe is it possible to rephrase it?
- Line 136-139: Sentence complicated to follow and a bit too long... maybe possible to shorten it? Or divide it into two?
- Line 139: qPCR was it mentioned before? Of not, please specify in full name before abbreviating. Even if it is described later on in the material and methods.
- Line 139: no comma between WT and *lox1*
- Line 142: which is not actually referring to induction, grammatically speaking, even if the sense comes through
- Line 133: HpODE is never reported in full form, only abbreviated
- Line 147-148: the complete name of a compound should appear only the first time in the text, then the abbreviation alone can be used. The first appearance of HODE is in the introduction
- Line 149: "classify LOX1 as a Fe-dependent 13-LOX" I believe it could be my flaw not fully understanding this classification... does it mean that LOX1 is the ortholog of 13-LOX? As in the introduction there is no specific mention of this LOX in particular, would it be possible to explain a bit better, for non-expert?
- Line 205-209: I understood after reading a couple of time, but this sentence is quite difficult to follow
- Line 233 - Fig S11: Clearly dealing with big datasets out of a considerable RNAseq analysis brings no little challenges, and I understand one analysis line for the enrichment had to be chosen above others (I would have still found cross enrichment quite interesting though). However, for a better representation of the metabolic processes enriched in one or the other strain I would have suggested to provide the enrichment ratio (number of DEGs in one process over the total number of genes classified in such process) rather than the pure count itself. I believe the enrichment ratio to be more informative.
- Line 483: while it is mentioned that the mutants were confirmed by Southern blot, the procedure and specific steps involved in this confirmation and the results could be included in the methods or at least in the supplementary information. How was the gDNA extracted?
- Lines 467-476: details on the RNA extraction are missing, as well as on the reverse transcription process and qPCR
- Would it be possible to increase a bit the character size in figure S3-S5-S8-S11-S12?

Major concerns:

- Line 238: here is only the first time that the reference to a gene in *T. atroviride* appear as Tatro. Where are these IDs from? In the material and methods, I could only see mentioned the IMI strain as wt. In such case, the new nomenclature is Triat2. In all the dataset provided there is only the uniprot code, no ID for the gene. How are the genes mentioned in the paper identifiable if no general ID, like the one provided in JGI, is given?
- Line 529: I am not aware of a KEGG annotation for *T. atroviride*. So, ok using clusterProfiler, but how was such analysis performed? Against which database?
- Finally, despite the quite considerable amount of data and results provided, I do not think that all the conclusions written in the discussion part are well supported.

Reviewer #2 (Public repository details (Required)):

All data were submitted to public repositories as stated in the Data availability section of the manuscript.

Reviewer #2 (Comments for the Author):

In this manuscript the authors describe the role of *lox1* and *plp1* in injury response of *Trichoderma atroviride*, also based on transcriptome analysis. They further reveal their role in production of the metabolite 13 HODE, which is associated with wound response. The study is very timely and interesting and the experimental design is sound. However, the language of the manuscript needs improvement.

Prior to publication there are several points that should be taken care of:

An introductory sentence in the abstract would be beneficial.

Please explain all abbreviations at first use in the text and also in the abstract. For example PUFA in the abstract or later on „ai“ which could mean after inoculation or after injury - should be clearly specified.

Especially in abstract and introduction, there are numerous grammar errors or awkward use of vocabulary. Please revise carefully throughout the manuscript or have the work edited by a native speaker. Generally use shorter sentences, which may also help to avoid grammar mistakes and to improve clarity.

First paragraph of results, lines 119-135: Where are these results shown? Specify with the description, where the figure for the results is or add a citation.

Throughout the manuscript it is sometimes unclear, which results were done by the authors in this study and which findings the cite from previous work. Please revise the wording accordingly, add references to figures or data and add citations where other work is described.

Line 147: Please write names of chemicals in english.

Line 194: Please describe how „regeneration efficiency“ is defined. Do only 50 % of hyphae elongate again, are they growing more slowly etc.?

Line 208: I think figure S9 would be better shown in the main manuscript, not supplementaries.

When describing overrepresented GO-terms, please add p-values and p-value threshold for statistical significance. Also, I think these results would be better shown in a list along with number of genes and p-values.

Since tmks, lox1 and plp1 all impact injury response indirectly, the authors should also discuss regulation of transcription factors separately in one section. Which TFs are regulated and in which way? Are there consistencies between the conditions they checked?
Additionally, discuss possible output pathways in more detail.

Line 322f.

Since the biosynthetic pathway of 13 HODE is obviously known in other organisms, please provide an overview on whether there are homologues of the biosynthetic genes in *T. atroviride* and how they are regulated in the presented transcriptomes. A figure would be helpful for that as well.

Line 383

Please provide p-value and ID for the homeodomain in this predicted protein.

Line 477

Were multiple independent mutants investigated per deletion or were re-transformants checked? Relevance of the mutations should be confirmed and this test specified in the text.

line 516

RNA was checked with a DNA chip. Is this a typo? Otherwise please explain why this works and provide a reference for the respective test.

Line 531

What are KO terms?

Figure 5 A, B

These chord diagrams look fancy, but with respect to information they are rather confusing. The authors should consider presenting these results in a different way.

Figure 5C

Error bars are missing here. Please add them.

Staff Comments:

Preparing Revision Guidelines

To submit your modified manuscript, log onto the eJP submission site at <https://spectrum.msubmit.net/cgi-bin/main.plex>. Go to Author Tasks and click the appropriate manuscript title to begin the revision process. The information that you entered when you

first submitted the paper will be displayed. Please update the information as necessary. Here are a few examples of required updates that authors must address:

Please return the manuscript within 60 days; if you cannot complete the modification within this time period, please contact me. If you do not wish to modify the manuscript and prefer to submit it to another journal, please notify me of your decision immediately so that the manuscript may be formally withdrawn from consideration by Microbiology Spectrum.

Dear Editor,

We wish to thank the reviewers for their constructive comments, which will certainly make our manuscript better. Below you will find a point by point rebuttal to the reviewer's comments.

Responses to Reviewer #1:

Q1. Reviewer #1 (Public repository details (Required)):

In the manuscript is written that that all the RNAseq data relevant for this analysis are available at NCBI omnibus. However the link, for understandable reason, is protected till defined date. But how are the reviewers able to verify the analysis if we do not have access to the data? Moreover, at the beginning of the results section, the authors refer to a previous RNAseq analysis, from a 2012 publication. When I tried to access those dataset, the link were giving an error. Of course this is not the authors fault, but considering some of the authors are the same between the two publication... it just doesn't give me a good feeling not being able to verify the original data

R1. The RNAseq data used in this manuscript are available for reviewers. To access the RNAseq data in Gene Expression Omnibus (<https://www.ncbi.nlm.nih.gov/geo/>), it is important to use the accession key: GSE235273 and secure token: ihghkyaqphyxzaj (write the secure token in the box in the section that reads as follows: "If you are a reviewer, enter secure token here"). On the other hand, the data from Hernández-Oñate et al. 2012 are no longer available for unknown reasons. We are investigating what happened with this information.

Q2. The manuscript "LOX1 and PLP1 dependent transcriptional reprogramming is essential for injury induced conidiophore development in a filamentous fungus" is a well written work, providing a lot of information and data regarding mechanical injury response in *Trichoderma atroviride*. Initially I identified only minor points that could be slightly changed to improve the manuscript. At a deeper look, however, some more relevant concerns were also raised.

R2. Thank you for your constructive comments. Below, we provide an answer to each one of them.

Minor concerns:

Q3. Line 112: 13-HODE the explicit form of the abbreviation is only written in the abstract. Wouldn't be more correct to have it at least once in the main text?

R3. Thank you for pointing this out. The reviewer is correct; we have spelled out 13-HODE in the abstract and main text. The revised text reads as follows in lines 37 and 114:

“13-hydroxy-9Z,11E-octadecadienoic acid (13-HODE)”

Q4. Line 130: The sentence starting with "While...", perhaps style choice, but just the fact that it started with that made the sentence difficult for me to understand at a first reading. Maybe is it possible to rephrase it?

R4. Thank you for mentioning this. For clarity, we have rewritten several lines in that paragraph, which included the elimination of the conditional connector “While”; see lines 130-136 of the new version of the manuscript. The revised text reads as follows:

*“We observed that treatment with eATP, which induces conidiation (2), increased the levels of *plp1* and *lox1* mRNA, those of *lox1* being even higher than upon mechanical injury (Fig. S1 A, B). In contrast, injury in the presence of BAPTA, a calcium-chelating agent that blocks regeneration and conidiation (21), decreased mRNA levels of both genes (Fig. S1 C). In the $\Delta tmk1$ and $\Delta tmk3$ mutants, affecting regeneration and injury-induced conidiation (2), respectively, injury resulted in upregulation of *plp1* and *lox1* but to lower levels than in the WT strain (Fig. S1 D, E).”*

Q5. Line 136-139: Sentence complicated to follow and a bit too long... maybe possible to shorten it? Or divide it into two?

R5. Thank you for this observation. The reviewer is correct; we have shortened the sentence to make it easier to follow. The revised text reads as follows in lines 140-142:

*“To determine when these genes reach their maximum expression level and whether there was cross-regulation between them, we analyzed by time course the expression of *lox1* and *plp1* in the WT strain and the $\Delta lox1$ and $\Delta plp1$ mutants (Fig. S2).”*

Q6. Line 139: qPCR was it mentioned before? Of not, please specify in full name before abbreviating. Even if it is described later on in the material and methods.

R6. Thank you for pointing this out. We have spelled out RTqPCR in the main text and material and methods. The revised text reads as follows in lines 143 and 481:

“Reverse transcription quantitative polymerase chain reaction (RT-qPCR) results showed that in the WT...”

Q7. Line 139: no comma between WT and *lox1*

R7. Thank you for pointing this out. We have deleted the comma. The revised text reads as follows in line 142-144:

“Reverse transcription quantitative polymerase chain reaction (RT-qPCR) results showed that in the WT lox1 increased its level of expression at 30 min after injury (ai) ...”

Q8. Line 142: which is not actually referring to induction, grammatically speaking, even if the sense comes through.

R8. Thank you for this observation; the word “induction” was incorrectly used. We have rephrased the sentence. The corrected text reads as follows in the lines 145-147:

“In comparison, plp1 reached a maximum expression at 30 min, showing a decrease at 90 min and a further decrease at 4 h (Fig. 1A).”

Q9. Line 133: HpODE is never reported in full form, only abbreviated

R9. Thank you for pointing this out. We have spelled out 13-HpODE and 9-HpODE. The revised text reads as follows (see lines 148-150):

“13-hydroperoxy-9Z,11E-octadecadienoic acid (13-HpODE) or 9-hydroperoxy-10E,12Z-octadecadienoic acid (9-HpODE)”

Q10. Line 147-148: the complete name of a compound should appear only the first time in the text, then the abbreviation alone can be used. The first appearance of HODE is in the introduction.

R10. Thank you for pointing this out. Instead of using the complete name of 13-HODE, we replaced it with the abbreviation. The revised text reads as follows in line 152:

“LOX1 and PLP1 work together to produce 13-HODE”

Q11. Line 149: "classify LOX1 as a Fe-dependent 13-LOX" I believe it could be my flaw not fully understanding this classification... does it mean that LOX1 is the ortholog of 13-LOX? As in the introduction there is no specific mention of this LOX in particular, would it be possible to explain a bit better, for non-expert?

R11. As suggested by the reviewer, we have written a brief overview of the type of enzyme LOX1 is and its main enzymatic characteristics; the same for PLP1. The revised text reads as follows (see lines 153-158):

“Using in silico analyses, we classified LOX1 as a non-heme oxygenase that uses Fe as a cofactor and inferred that it oxidizes 13-C of linoleic acid (LA) to produce 13(S)-HpODE (Fig. S3-4). LOX1 is closely related to human 15-LOX and Arabidopsis thaliana 9- or 13-LOX. PLP1, on the other hand, is a functional small acyl hydrolase with a

phosphate-binding motif related to a human PLP (PNPLA8) and three A. thaliana PLP (AtPLA-IVA and AtPLA-IVB and AtPLA-IIA) (Fig. S5)."

Q12. Line 205-209: I understood after reading a couple of time, but this sentence is quite difficult to follow

R12. We have rephrased a couple of sentences to make it easier to follow. The revised text reads as follows (see lines 215-219):

"From the differential gene expression analysis, we found that 46.24% and 51.67% of the genes with altered expression patterns relative to the WT at 90 min and 4 h ai, respectively, had the same behavior in the two mutants (Fig. 4B). Similarly, the mutants shared 61.22% of the genes with altered expression patterns in the non-injured control (Fig. 4B)."

Q13. Line 233 - Fig S11: Clearly dealing with big datasets out of a considerable RNAseq analysis brings no little challenges, and I understand one analysis line for the enrichment had to be chosen above others (I would have still found cross enrichment quite interesting though). However, for a better representation of the metabolic processes enriched in one or the other strain I would have suggested to provide the enrichment ratio (number of DEGs in one process over the total number of genes classified in such process) rather than the pure count itself. I believe the enrichment ratio to be more informative.

R13. As suggested by the reviewer, we are now providing the enrichment ratio (Gene Ratio) rather than the count.

Q14. Line 483: while it is mentioned that the mutants were confirmed by Southern blot, the procedure and specific steps involved in this confirmation and the results could be included in the methods or at least in the supplementary information. How was the gDNA extracted?

R14. As suggested by the reviewer, we have described the procedure for Southern blot and DNA extraction in the supplementary information (section 1.1).

Q15. Lines 467-476: details on the RNA extraction are missing, as well as on the reverse transcription process and qPCR

R15. As suggested by the reviewer, we have described the procedure for RNA extraction and qPCR in Supplementary Information (section 1.2).

Q16. Would it be possible to increase a bit the character size in figure S3-S5-S8-S11-S12?

R16. As suggested, we have increased the character size in the supplemental figures.

Major concerns:

Q17. Line 238: here is only the first time that the reference to a gene in *T. atroviride* appear as Tatro. Where are these IDs from? In the material and methods, I could only see mentioned the IMI strain as wt. In such case, the new nomenclature is Triat2. In all the dataset provided there is only the uniprot code, no ID for the gene. How are the genes mentioned in the paper identifiable if no general ID, like the one provided in JGI, is given?

*R17. Thank you for mentioning this. We used the *Trichoderma atroviride* genome assembled in our lab for mapping sequencing reads. The genome sequence is available on NCBI genomes, accession number: JAEAGS000000000. For reasons of novelty and logistics, this genome uses an ID for each gene model different from that of the genome available from the JGI. Since this genome contains more annotated genes than the Triat2 version, several new genes do not have an equivalent JGI ID, so we decided not to use the Triat2 version's IDs. In column 7 of datasets (DS1, DS6, and DS7), we mention the corresponding JGI's IDs for each gene model, except for new gene models (previously undetected).*

Q18. Line 529: I am not aware of a KEGG annotation for *T. atroviride*. So, ok using clusterProfiler, but how was such analysis performed? Against which database?

R18. Thank you for pointing this out. To do KEGG analysis in non-model organisms, especially those organisms not listed in the KEGG database (http://www.genome.jp/kegg/catalog/org_list.html), we used `enrichKEGG` function from `clusterProfiler` package, specifying `organism = "ko"`. `ko` is defined as "reference pathways maps linked to KO entries (KO = K number, representing a functional ortholog that corresponds to a KEGG pathway node)." Briefly, the `enrichKEGG` function connects with KEGG pathways to perform the enrichment. First, we obtained the K number from functional annotation with `eggNOG-mapper` (<http://eggnog-mapper.embl.de/>). Then, we used the K number mapping to the `ko` pathway online for enrichment analysis. We have added a link to access the script used for the enrichment analysis in this work in lines 551-552. The text reads as follows in materials and methods: "The scripts used for the enrichment analysis in this work are available at https://github.com/OrlanC/cp_enrichment."

Q19. Finally, despite the quite considerable amount of data and results provided, I do not think that all the conclusions written in the discussion part are well supported.

R19. Unfortunately, it is impossible to answer your concerns directly without reference to specific conclusions. However, we firmly believe that all our conclusions are supported by our data and by previous reports on other organisms. Our main conclusions are:

- 1) LOX1 and PLP1 work together to produce 13-HODE.*
- 2) LOX1 and PLP1 regulate injury-induced conidiation.*
- 3) LOX1 and PLP1 have a similar transcriptional response after injury required for activating essential physiological and molecular mechanisms in Trichoderma atroviride to induce conidiation.*

According to reports in other ascomycetes, we also discussed the mechanisms of differential genes found in our analysis that could have a role in injury-induced conidiation. However, we did not conclude; we instead suggested the possible function of these genes in response to wounding in T. atroviride.

In any case, we revised the discussion and rephrased the last paragraph to avoid overstatement. The text reads as follows (lines 439-448):

“Our research has demonstrated two important findings. First, regenerated hyphae require LOX1 and PLP1 to produce 13-HODE at significant levels. Second, LOX1 and PLP1 regulate injury-induced conidiation through transcriptional reprogramming that triggers activation of biochemical processes, induction of TFs, and structural proteins involved in aerial hyphae and asexual development in ascomycetes (Fig. 8). These results suggest that enzymatically oxidized lipids are a new input signal on damage-sensitized cells that induces injury-induced conidiation, acting downstream at different cellular levels and rapidly activating a transcriptional program of asexual development. Thus, we propose that LOX1 and PLP1 work together to produce LOX-initiated oxidized PUFAs in fungi that induce asexual development to ensure survival in harmful environmental conditions.”

Responses to Reviewer #2

Public repository details (Required):

All data were submitted to public repositories as stated in the Data availability section of the manuscript.

Q1. In this manuscript the authors describe the role of lox1 and plp1 in injury response of *Trichoderma atroviride*, also based on transcriptome analysis. They further reveal their role in production of the metabolite 13 HODE, which is associated with wound response. The study is very timely and interesting and the experimental design is sound. However, the language of the manuscript needs improvement.

R1. Thank you for your positive comments. We carefully revised language use and had the manuscript corrected by a native English speaker.

Q2. An introductory sentence in the abstract would be beneficial.

R2. Thank you for the suggestion. We have added an introductory sentence to the abstract.

Q3. Please explain all abbreviations at first use in the text and also in the abstract. For example PUFA in the abstract or later on „ai" which could mean after inoculation or after injury - should be clearly specified.

R3. We agree with the reviewer's assessment. Accordingly, throughout the manuscript, we have spelled out all abbreviations at first use.

Q4. Especially in abstract and introduction, there are numerous grammar errors or awkward use of vocabulary. Please revise carefully throughout the manuscript or have the work edited by a native speaker.

Generally use shorter sentences, which may also help to avoid grammar mistakes and to improve clarity.

R4. We carefully revised language use and had the manuscript corrected by a native English speaker. We paid special attention to the abstract and introduction.

Q5. First paragraph of results, lines 119-135: Where are these results shown? Specify with the description, where the figure for the results is or add a citation.

R5. Thank you for pointing this out; we have added the cross-reference to Supplementary Figure 1, showing the results. The changes are in the revised text (lines 132, 134 & 136), Supplementary Figure 1, and the legend to Supplementary Fig. 1.

Q6. Throughout the manuscript it is sometimes unclear, which results were done by the authors in this study and which findings they cite from previous work. Please revise the

wording accordingly, add references to figures or data and add citations where other work is described.

R6. As suggested by the reviewer, for clarity, we have carefully revised citations and, where necessary indicated the published reference or figure and dataset derived from the present study to which the text refers.

Q7. Line 147: Please write names of chemicals in english.

R7. Thank you for pointing this out. We have revised that chemical names are correctly written throughout the manuscript and modified them when required.

Q8. Line 194: Please describe how „regeneration efficiency" is defined. Do only 50 % of hyphae elongate again, are they growing more slowly etc.?

R8. As suggested by the reviewer, we have described regeneration capacity in the text (lines 199-200) as "We quantified new hyphal tips and thin hyphae emergence near the rupture point of damaged hyphae over time (regeneration)."; Efficiency is defined as the percentage of hyphae that regenerate of the total damaged hyphae observed.

Q9. Line 208: I think figure S9 would be better shown in the main manuscript, not supplementaries.

R9. The new version of the manuscript now includes the former Figure S9 as Figure 4B.

Q10. When describing overrepresented GO-terms, please add p-values and p-value threshold for statistical significance. Also, I think these results would be better shown in a list along with number of genes and p-values.

R10. We agree with the reviewer's assessment. Throughout the manuscript, we have added p-values and p-values thresholds. On the other hand, the lists of enrichment analysis are in Data Sets 2, 3, and 5.

Q11. Since tmks, lox1 and plp1 all impact injury response indirectly, the authors should also discuss regulation of transcription factors separately in one section. Which TFs are regulated and in which way? Are there consistencies between the conditions they checked? Additionally, discuss possible output pathways in more detail.

R11. Although this is an important suggestion, the regulation of some TF is still being determined; hence, discussing a possible regulation explanation would be speculative

*because no solid data would support it. Regarding the tmks, consistently with our data, Medina-Castellanos et al. (2018; <https://www.ncbi.nlm.nih.gov/pmc/articles/PMC6291166/>) reported a transcriptomic analysis of the response to injury of a $\Delta tmk3$ mutant, where they found that the mutation results in the downregulation of *brlA* and *azf1*. However, the transcriptional response to wounding in that study was analyzed 30 min after injury, while our data were collected 90 min and 4 h after injury. Therefore, the results cannot be directly compared.*

Q12. Line 322f. Since the biosynthetic pathway of 13 HODE is obviously known in other organisms, please provide an overview on whether there are homologues of the biosynthetic genes in *T. atroviride* and how they are regulated in the presented transcriptomes. A figure would be helpful for that as well.

R12. Thank you for pointing this out. We have revised the text and realized that we used incorrectly the phrase “lipoxygenase pathway” to indicate the link between patatin-like phospholipase and lipoxygenase. According to the literature’s LOX pathway definition, PUFAs oxidation is carried out first by the LOX step and subsequently by other (mainly enzymatic) reactions. However, until now, we only know about the LOX step in fungi. To avoid confusion and highlight that the activity of PLP1 is upstream to the LOX step, we remove the sentence “This suggests that they are components of the lipoxygenase pathway, like plants and animals” and replaced it with “These results suggest that PLP1 mediates PUFA levels to produce 13-HODE via the LOX1 step” see lines 341-342. Given this correction and continuing with the reviewer’s suggestion, instead of providing an overview of the lipoxygenase pathway, we have added in the manuscript an overview of how patatin-like phospholipase are connected to Lipoxygenase, as well as the function of their homologs in plants and animals (see lines 331-341).

Q13. Line 383. Please provide p-value and ID for the homeodomain in this predicted protein.

*R13. In the new version of the manuscript, we now provide the p-value and ID for the homeodomain of this protein (line 398). The revised text reads as follows: “Another TF induced only in the WT is the homeobox gene *Tatro_005212-T1* (JGI: 164928) (P-value = 0.00744, FDR < 0.05) ...”*

Q14. Line 477. Were multiple independent mutants investigated per deletion or were re-transformants checked? Relevance of the mutations should be confirmed and this test specified in the text.

R14. We checked independent lox1 and plp1 mutants (Supplementary Figure 7). The time course response to mechanical injury of mutants lox1-5 and plp1-9 was consistent with that of the lox1-6 and plp1-2 mutants used in all our experiments (Fig. S7A). Similarly, Speckbacher et al., (2020; <https://pubmed.ncbi.nlm.nih.gov/32973724/>) recently reported that lox1 mutants of another isolate of Trichoderma atroviride, T. atroviride P1, are affected in injury-induced conidiation, which is also consistent with the defects observed in the lox1 mutant generated in our study.

Q15. line 516. RNA was checked with a DNA chip. Is this a typo? Otherwise please explain why this works and provide a reference for the respective test.

R15. Thank you for pointing this out. The reviewer is correct; it was a typo. The “DNA chip” has been corrected (lines 531-532). The revised text reads as follows: “The quality and quantity of the total RNA were determined using an RNA 6000 Nano Chip run in an Agilent Bioanalyzer platform.”

Q16. Line 531. What are KO terms?

R16. We incorrectly used the “KO terms” to indicate KEGG pathways. This mistake has been corrected on line XXX and in the Supplementary Data Set 4 description. The KEGG Orthology (KO) system is the basis for linking genomic information to higher-level systemic functional information through KEGG PATHWAY mapping and BRITE mapping.

Q17. Figure 5 A, B. These chord diagrams look fancy, but with respect to information they are rather confusing. The authors should consider presenting these results in a different way.

R17. We appreciate the reviewer’s feedback. However, this plot is more informative since it depicts the linkage of genes and GO terms, unlike other diagrams that only represent GO terms.

Q18. Figure 5C. Error bars are missing here. Please add them.

R18. We added the missing error bars in the graphs shown in Figure 5C.

October 4, 2023

Dr. Alfredo H Herrera-Estrella
Centro de Investigacion y de Estudios Avanzados del Instituto Politecnico Nacional
National Laboratory of Genomics for Biodiversity-Advanced Genomics Unit
Km. 9.6 Libramiento Norte Carretera Irapuato-Leon
Irapuato, Guanajuato 36824
Mexico

Re: Spectrum02607-23R1 (LOX1 and PLP1-dependent transcriptional reprogramming is essential for injury-induced conidiophore development in a filamentous fungus)

Dear Dr. Alfredo H Herrera-Estrella:

I am happy to report to you that your manuscript has been accepted, and I am forwarding it to the ASM Journals Department for publication. You will be notified when your proofs are ready to be viewed.

Sincerely,

Lea Atanasova
Editor, Microbiology Spectrum
